# An integrative multiomic network model links lipid metabolism to glucose regulation in coronary artery disease

Ariella T. Cohain[1,12], William T. Barrington[2,12], Daniel M. Jordan [1,3,12], Noam D. Beckmann[1], Carmen A. Argmann [1], Sander M. Houten[1], Alexander W. Charney[1,4], Raili Ermel [5], Katyayani Sukhavasi [5], Oscar Franzen [6], Simon Koplev [1], Carl Whatling[7], Gillian M. Belbin[1,3], Jialiang Yang[1], Ke Hao[1], Eimear E. Kenny[1,3], Zhidong Tu[1], Jun Zhu [1], Li-Ming Gan [8], Ron Do [1,3], Chiara Giannarelli [1,9], Jason C. Kovacic [9], Arno Ruusalepp[5], Aldons J. Lusis [2], Johan L. M. Bjorkegren [1,10,12✉] & Eric E. Schadt [1,11,12✉]

Elevated plasma cholesterol and type 2 diabetes (T2D) are associated with coronary artery disease (CAD). Individuals treated with cholesterol-lowering statins have increased T2D risk, while individuals with hypercholesterolemia have reduced T2D risk. We explore the relationship between lipid and glucose control by constructing network models from the STARNET study with sequencing data from seven cardiometabolic tissues obtained from CAD patients during coronary artery by-pass grafting surgery. By integrating gene expression, genotype, metabolomic, and clinical data, we identify a glucose and lipid determining (GLD) regulatory network showing inverse relationships with lipid and glucose traits. Master regulators of the GLD network also impact lipid and glucose levels in inverse directions. Experimental inhibition of one of the GLD network master regulators, lanosterol synthase (*LSS*), in mice confirms the inverse relationships to glucose and lipid levels as predicted by our model and provides mechanistic insights.

[1] Department of Genetics and Genomic Science and Institute for Multiscale Biology, Icahn School of Medicine at Mount Sinai, New York, NY 10029, USA. [2] Department of Human Genetics/Medicine, David Geffen School of Medicine, University of California Los Angeles (UCLA), Los Angeles, CA, USA. [3] The Charles Bronfman Institute for Personalized Medicine, Icahn School of Medicine at Mount Sinai, New York, NY 10029, USA. [4] Department of Psychiatry, Icahn School of Medicine at Mount Sinai, New York, NY 10029, USA. [5] Department of Cardiac Surgery, Tartu University Hospital, Tartu, Estonia. [6] Integrated Cardio Metabolic Centre, Department of Medicine, Karolinska Institutet, Karolinska Universitetssjukhuset, Huddinge, Sweden. [7] Translational Science, Cardiovascular, Renal and Metabolism, IMED Biotech Unit, AstraZeneca, Gothenburg, Sweden. [8] Early Clinical Development, Cardiovascular, Renal and Metabolism, IMED Biotech Unit, AstraZeneca, Gothenburg, Sweden. [9] Cardiovascular Research Centre, Icahn School of Medicine at Mount Sinai, New York, NY 10029, USA. [10] Clinical Gene Networks AB, Stockholm, Sweden. [11] Sema4, Stamford, CT, USA. [12] These authors contributed equally: Ariella T. Cohain, William T. Barrington, Daniel M. Jordan, Johan L.M. Bjorkegren, Eric E. Schadt. ✉email: johan.bjorkegren@mssm.edu; eric.schadt@mssm.edu

ow-density lipoprotein (LDL) cholesterol has a well-established causal role in coronary artery disease (CAD), representing not only the most significant biomarker for CAD risk but also the primary therapeutic target for treatment and prevention. type 2 diabetes (T2D) is another well-established risk for CAD[1,2]. However, a perplexing juxtaposition between LDL and T2D remains to be fully resolved: patients with familial hypercholesterolemia (increased LDL) have a dramatically increased risk for CAD and a decreased risk for T2D[3]. In addition, lipid transport has been shown to be involved in regulating insulin secretion[4], while both phenome wide association studies (PheWAS) and Mendelian randomization (MR) approaches have found that single nucleotide polymorphisms (SNPs) associated with increased risk of CAD are associated with increased LDL levels, but decreased fasting insulin levels and T2D risk[5]. MR studies have also shown the inverse relationship with elevated LDL or triglyceride (TG) levels associated with a reduction in T2D risk[6–8].

The relationship between LDL, CAD, and T2D has also been well established pharmacologically. Lowering plasma LDL via lipid-lowering drugs not only results in a reduction in CAD-related clinical events and the lipid content of coronary atherosclerotic plaques[9], but can also increase T2D risk[10]. Numerous, large cohort studies have shown that statin therapies, while producing a significant cardiovascular benefit, also increase the risk of T2D[10–15]. Moreover, MR studies of SNPs proximal to *HMGCR*, the gene encoding the enzyme targeted by statins, suggest that increased risk of T2D noted with statins could at least partially be explained by HMGCR inhibition[16]. However, the multiple biochemical effects of statins have not been fully characterized at the molecular level, and thus, how statins elevate T2D risk is not presently understood[17]. Many studies have established the specific roles genes play in lipid and glucose metabolism[18–21]. At the genomic level, genetic variants that modulate these traits have been identified[22–24]. However, this canonical view fails to reveal the inter-metabolic regulatory control features that serve to establish the inverse relationship that exists between these traits. We hypothesize that comprehensive transcriptomic models that place lipid and glucose modulation in a broader molecular context are necessary to link canonical pathways that in turn can uncover the causes that underlie the seemingly paradoxical links between lipid levels, CAD, and T2D.

Here we employ a multiscale, integrative network approach using the Stockholm-Tartu Atherosclerosis Reverse Network Engineering Team (STARNET) study[25], which to date is the largest human dataset designed to study CAD and related cardiometabolic disorders in a multiomics and multi-tissue framework. We construct molecular network models across all seven tissues: aortic root (AOR), mammary artery (MAM), liver (LIV), subcutaneous fat (SF), visceral fat (VAF), skeletal muscle (SKLM), and whole blood (Blood) in STARNET to assess the molecular components of lipid and glucose metabolism. We discover a highly conserved liver subnetwork, referred to as the glucose and lipid determining (GLD) network, as the most coherent network model providing causal regulatory linkages defining an inverse relationship between lipid and glucose metabolism.

## Results

We pursued a comprehensive data-driven computational strategy to better elucidate the molecular control of plasma cholesterol and glucose metabolism in the context of CAD, given that CAD provides a more extreme context in which metabolic traits such as cholesterol levels are more significantly perturbed, leading to clinical manifestation of CAD and requiring surgical intervention. Our principal data resource was STARNET, a cohort recruited at

the Tartu University hospital in Estonia including patients diagnosed with CAD who were eligible for coronary artery by-pass grafting (CABG) surgery. We analyzed the extensive omic datasets provided in the STARNET cohort, including genotypes, gene expression levels from seven arterial wall, blood, and metabolic tissues, which is combined with a variety of clinical data that in addition to age, gender, body-mass index (BMI) and ethnicity also include standard blood biochemistry including plasma lipids and blood glucose levels, a full Nightingale metabolomic profile[26–28], and additionally present and previous diagnoses and medications. A more detailed profile of this cohort has been published previously[25].

To uncover the molecular drivers of plasma LDL and blood glucose levels, we employed an integrative network approach utilizing multiple different data captures as depicted in Fig. 1. The overall goals were to systematically characterize all gene-clinical trait associations, all clusters of coexpressed genes, and the associations of those gene clusters with clinical trait data and metabolomics. To achieve this, we performed a series of three analyses. First, we performed differential expression analysis using STARNET gene expression data across seven tissues and clinical information, identifying a total of 12,872 unique genes across seven tissues whose expression was correlated with at least one glucose or lipid related clinical trait. Second, we performed coexpression clustering analysis on all 7 tissues, identifying 140 tissue-specific gene clusters. Comparing these clusters to the results of the differential expression analysis, we found a total of 20 clusters of coexpressed genes across three tissues enriched for these glucose and lipid correlated genes. We identified one of these modules in particular as a liver glucose and lipid determining (GLD) module and verified its presence in two additional human transcriptomics datasets and one mouse transcriptomics dataset. Finally, we constructed four different probabilistic causal network models integrating transcriptomics, genomics, clinical traits, and metabolomics. We then mined all four of these networks to identify 30 candidate master regulator genes for the GLD module, and validated the physiological relevance of the top candidate with respect to plasma lipid and glucose control using an in vivo experimental mouse model system.

The first step in our integrative analysis process was to construct tissue-specific differential expression (DE) signatures correlated with traits by identifying genes whose expression was significantly correlated with the clinical measurements we had of plasma lipids (total plasma cholesterol, LDL, HDL, and triglycerides), diagnosis of hyperlipidemia, active statin therapy, blood glucose traits (plasma HbA1c, blood glucose, and insulin), diagnosis of T2D, and active oral anti-diabetic therapy. The liver tissue provided the largest trait-correlated DE signature, comprising a total of 4653 unique genes (false discovery rate, FDR, <5%; Fig. 2A and Supplementary Data 1). We ran gene ontology (GO) enrichment analysis separately for each trait-correlated DE signature in each tissue to assess their biological relevance. The glucose and lipid trait-correlated DE signatures had highly overlapping GO terms across all traits and all tissues, with 586 GO terms shared in common across all. This represents a significant sharing of GO terms compared to what would be expected for independent lists (odds ratio (OR): 108.8, Fisher's Exact Test $P < 3E\text{-}324$, one-tailed), reflecting a high degree of interaction between pathways associated with lipid and glucose biology across tissues (Supplementary Fig. 1A and Supplementary Data 2). For example, in liver, where the expectation had been that lipid traits would be related to cholesterol biosynthesis and glucose traits related to pyruvate metabolism, but not necessarily to both, we observed that pathways enriched for both lipid and glucose-related traits included biosynthesis of sterols and pyruvate metabolism (Supplementary Fig. 1B).

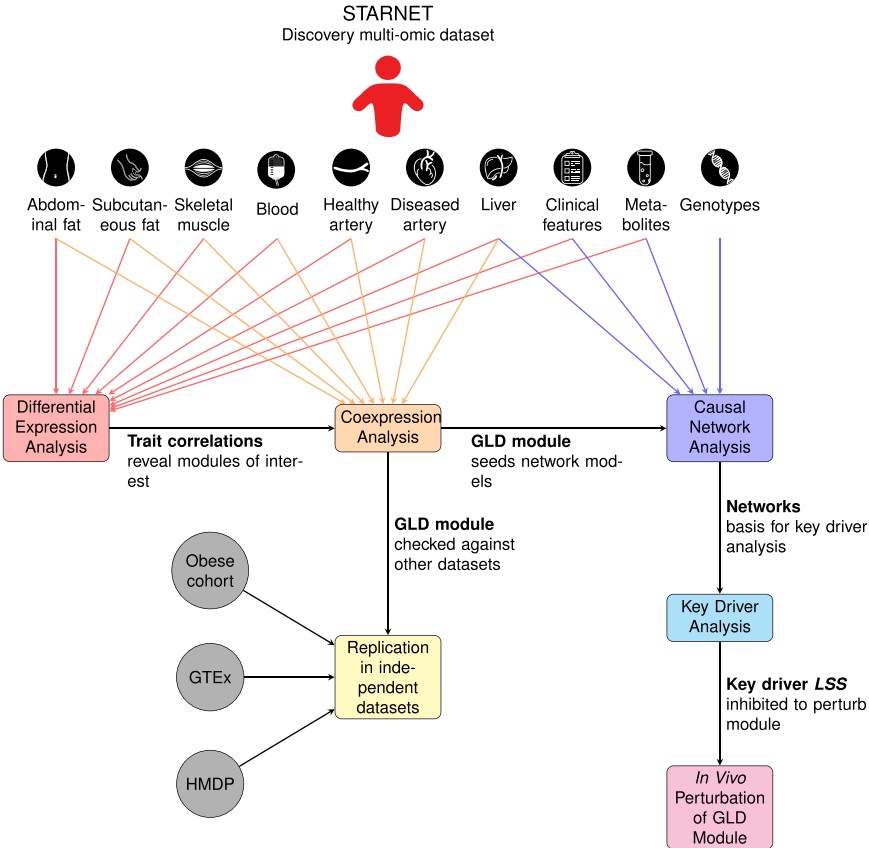

**Fig. 1 Schematic overview of workflow.** We first conducted differential expression analysis to identify genes whose expression is correlated with glucose and lipid traits. We then used these trait correlations in combination with a coexpression analysis to identify a GLD module that simultaneously regulates both glucose metabolism and lipid metabolism. We replicated this analysis in a mouse model and two additional human cohorts and found equivalent modules in each. We performed causal inference and key driver analysis on the GLD module to identify master regulator genes including *LSS*. Finally, we inhibited Lss in a mouse model to demonstrate the function of the GLD module.

**Identification of glucose and lipid metabolism coexpression modules**. While trait-correlated DE signatures broadly identify genes associated with lipid and glucose traits, they do not capture the rich correlation structure of biological processes, which are captured with high statistical power in the STARNET data. Coexpression network analysis provides a data-driven approach to identify co-regulated sets of genes across the STARNET cohort. To characterize the correlation structure among genes comprising the different trait-correlated DE signatures within the STARNET data, we applied Weighted Gene Coexpression Network Analysis (WGCNA), a method that organizes all pairwise correlations among genes to identify groups of genes that are coexpressed (modules)[29–33]. We identified a total of 140 modules across all seven tissues (Supplementary Data 3). In all, 20 of these modules in the liver (N = 6), VAF (N = 5), and SF (N = 9) were enriched for both lipid and glucose correlated DE signatures (Supplementary Fig. 2). In all, 16 of these modules had statistically significant correlations of any magnitude with plasma lipid and blood glucose levels when considering the first principal component of gene expression in each module (mean variation explained: 52%, range: 42–64%, Supplementary Table 1). Similar but weaker correlations were observed in higher principal components (Supplementary Data 4).

Though no principal component of any module showed overwhelmingly strong correlations with lipid or glucose traits, only one module in the liver (hereafter, the glucose and lipid determining module; GLD) had any principal component that was negatively correlated with lipid traits (LDL and plasma cholesterol levels) and positively correlated with glucose traits (HbA1c and blood glucose levels; FDR < 5%, Fig. 2). The GLD module was significantly enriched for the KEGG pathways Steroid Biosynthesis (fold enrichment: 190.8, Fisher's Exact Test FDR: 7.18E-24, one-tailed) and Terpenoid Backbone Biosynthesis (fold enrichment: 127.2, Fisher's Exact Test FDR: 1.75E-11, one-tailed; Supplementary Fig. 3 and Supplementary Data 5 for full results). We further investigated the metabolic functions of GLD through a metabolomic panel consisting of 233 higher-resolution lipid and glucose traits measured in STARNET[25] using nuclear magnetic resonance (NMR) spectroscopy[26–28]. The first principal component of GLD module gene expression had weak but significant correlations with 113 metabolites (FDR < 5%, Supplementary Data 7 and Supplementary Fig. 4), including a positive correlation with Glucose (Glc) and negative correlations with the cholesterol ester component of many lipoprotein size fractions (VLDL: XS, S, M, L, XXL, LDL: S, M, L, HDL: S, M, L, and IDL), consistent with the associations between this module and the glucose and lipid clinical traits (Fig. 3A).

**GLD replicates in non-CAD human livers and is conserved in mice**. To assess the robustness of the GLD network module in independent human studies and across species in mouse, we reconstructed coexpression networks from cohorts independent of STARNET. In a previously described cohort of 600 morbidly obese patients undergoing bariatric bypass surgery[34], three tissues (liver, subcutaneous fat, and visceral fat) were profiled using a

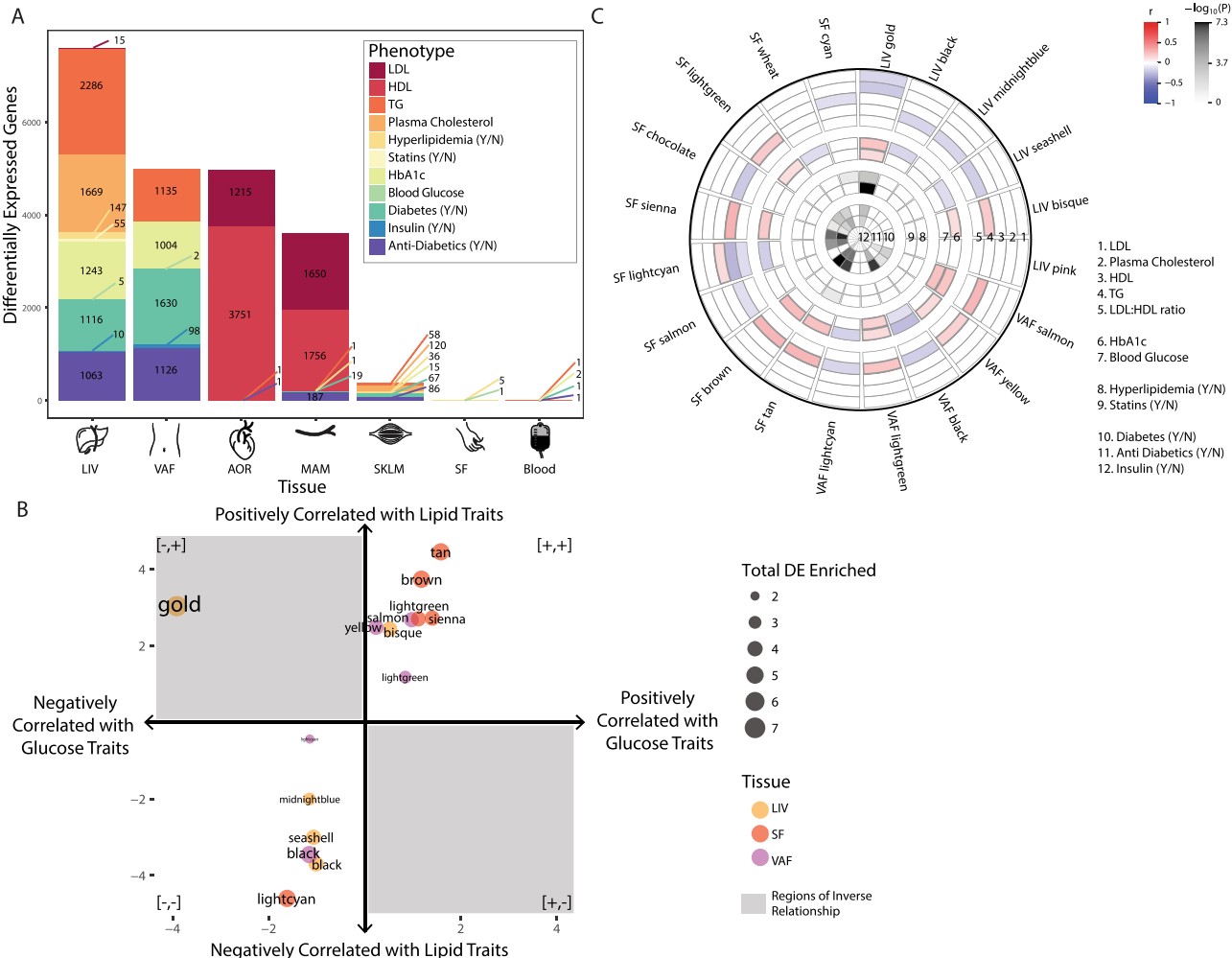

**Fig. 2 Differential expression analysis shows genes and modules associated with glucose and lipid traits. A** The number of trait-correlated differentially expressed (DE) genes at an FDR ≤5% for each tissue and clinical traits related to lipid and glucose metabolism. The x-axis displays the tissues and y-axis is a total of all the different signatures. **B** The 20 modules which were enriched for lipid and glucose trait-correlated DE signatures are shown, where the x-axis and y-axis correspond to the number of lipid and glucose traits, respectively and the direction of correlation of the module's 1st PC with the traits. Point size and text represents the total number of signatures the module is enriched for and color represents the tissue the module is found in. The areas of inverse relationship are shown in a gray box. **C** Correlation of the 20 modules' 1st PC with the glucose & lipid clinical traits, color represents the direction of correlation (only FDR ≤ 5% are depicted).

custom gene expression microarray[34]. We constructed coexpression networks for each tissue, identifying 40, 56, and 32 modules in liver, subcutaneous fat, and visceral fat, respectively. We tested whether the GLD module genes were enriched in any of these 128 modules, and observed only a single module in liver (39 probes, 34 genes) with strong overlap (OR: 2510.8, Fisher's Exact Test FDR: 1.04E-70, one-tailed). This is the largest overlap observed between any module in the obese cohort and any module in STARNET liver (Supplementary Data 8). Consistent with observations from STARNET, correlating the first principal component of this liver module with clinical metabolic traits measured in these 600 individuals found statistically significant positive correlations with glucose levels and a statistically significant negative correlation with plasma LDL levels (FDR < 5%; Supplementary Table 2).

We next sought to examine the GLD network genes from STARNET across a broader array of tissues collected from individuals in the GTEx study[35]. While we did not have access to clinical data from the GTEx study, the 12 tissues currently represented in GTEx with sufficient sample sizes to construct coexpression networks enabled the examination of the coherence

of the GLD network genes across multiple tissues. In order to partially control for the lack of individual phenotype data in GTEx, we split the GTEx cohort into two groups based on age, so that the younger group would be more likely to be heart-healthy (see Methods section). After splitting in this way, we built coxpression networks for each group across all tissues. We identified enrichments of the GLD network genes in modules from the esophagus, transformed fibroblast, lung, aorta, thyroid, tibial nerve, sun exposed skin, skeletal muscle, and subcutaneous adipose tissues (Supplementary Table 3), but we found that the GLD module was most highly conserved in liver coexpression modules from both age groups (35/47 and 30/36 genes were found in the original GLD, OR: 3368 and 5172, Fisher's Exact Test one-tailed FDR: 1.86E-88 and 1.15E-77 for old and young, respectively).

We then sought data from another species than human preferably mouse to examine to what extent the GLD module is evolutionary conserved and to enable subsequent experimental validation of the GLD module. We identified the hybrid mouse diversity panel (HMDP)[36]. The HMDP represents not only one but a diverse panel of over 100 mouse strains which makes the

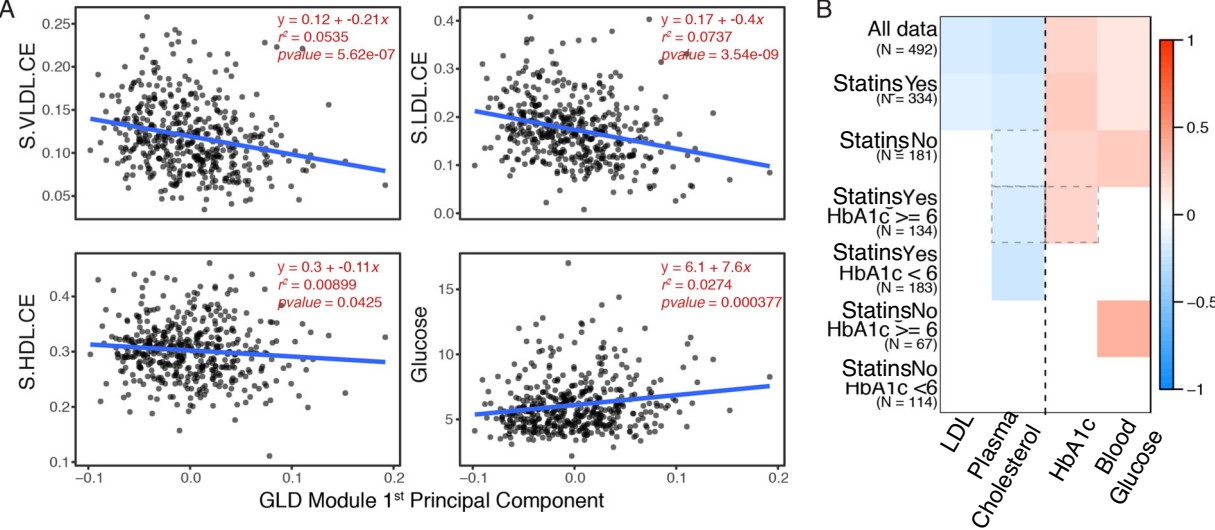

**Fig. 3 Association of GLD module gene expression with glucose and lipid traits. A** Correlation plots for cholesterol ester component of VLDL, LDL, HDL (particle size Small), and Glucose (Glc) with the GLD module 1st PC. Linear regression line is shown in blue with equation in red. **B** GLD module across all individuals, those split based only on statins and those split by statins and HbA1c levels. Each module's 1st PC was correlated with the clinical traits of interest (shown on the *x*-axis). Correlations in a dashed gray box represent FDR < 10%, all other correlations have an FDR <5%.

data more reflective in capturing natural diversity. In addition to data from the wild-type chow fed HMDP mice, we focused on the HMDP strains that have been bred onto an Apoe Leiden genetic background providing a human-like dyslipidaemia making these mice more susceptible to atherosclerosis. Looking at the liver expression values for these two sets of HMDP mice, we found that the GLD gene mouse orthologues were positively correlated with blood glucose and some genes that were negatively correlated with cholesterol (Supplementary Fig. 5).

**Association between therapeutic drugs and the GLD module.** Given the well-established pharmacologic effect of cholesterol-lowering medications such as statins to simultaneously increase glucose levels, and given that the GLD module was found associated with the diagnoses of hypertension and diabetes, as well as medication with statins, oral anti-diabetics, and beta-blockers (FDR < 5%; Fig. 2C), we explored whether the GLD module and its association with glucose and lipid traits were driven by patients who were taking these medications. We divided the STARNET cohort into two groups: one comprised of individuals reported to take statins (N = 334, from here on referred to as the statin group), and one of individuals not reported to take any lipid-lowering drugs (N = 182, from here on referred to as the no statin group). We then separately constructed coexpression network modules for each of these two groups (N = 32 in the statin group and N = 22 in the no statin group). We found that the original GLD module was conserved only in one specific module in both groups, with a 90% gene overlap in the statin group (OR: 15113, Fisher's Exact Test FDR: 3E-140, one-tailed) and a 92.5% overlap in the no statin group (OR: 11523, Fisher's Exact Test FDR: 2E-126, one-tailed). The first principal component of the GLD equivalent module in the statin group, explaining 54.4% of its variation, was positively correlated with HbA1c and blood glucose and negatively correlated to total and LDL plasma cholesterol levels at an FDR < 5% (Fig. 3B and Supplementary Data 6). The first principal component of the GLD equivalent module in the no statin group, also explaining 54.5% of its variation, was again positively correlated to HbA1c and blood glucose (FDR < 5%) but the negative correlation to total plasma cholesterol was only significant at FDR < 10% and the plasma

LDL correlation was lost (Fig. 3B). In the metabolomics data, 110 features were significantly different (FDR < 0.05) between STARNET individuals in the statin versus the no statin groups (Supplementary Data 7). Of these 110 features, 64 cholesterol, lipoprotein, and fatty acid measurements coincided with those correlated to the GLD module (inferred from all individuals), representing a significant overlap (OR: 2.09, Fisher's Exact Test *P*: 0.0038, one-tailed).

To assess whether the structure of the GLD module may be affected by statins or HbA1c levels, we further split the two groups defined above based on high and low HbA1c levels (≤6 representing the low HbA1c group and >6 the high group). Given that the smallest sample size across the four resulting groups was 67 (in the no statin and high HbA1c group), we had sufficient sample sizes to produce robust coexpression networks for each group. Across the four coexpression networks, only one module for each network significantly overlapped the original GLD module, indicating that under different drug and HbA1c conditions, the GLD module is highly conserved, and thus the correlation structure reflected in this module cannot solely be driven by statins or HbA1c levels. The GLD module correlations to the lipid and glucose traits were also largely conserved across the different groups. For example, on the statin group with low HbA1c levels the GLD module had a significant negative correlation to plasma cholesterol (FDR < 5%), whereas in the no statin group with high HbA1c the GLD module was positively correlated with blood glucose (Fig. 3B).

**Fine-mapping genetic risk loci for metabolic traits with GLD module genes.** Leveraging DNA information allows us to determine causality, as we know that DNA must be causally upstream of gene expression – that is, a genetic variant can cause altered gene expression but a change in gene expression cannot modify germ line DNA. We started by assessing expression quantitative trait loci (eQTL) in STARNET liver samples. At an FDR < 5% we detected 7010 genes containing an eQTL (46.67% genes in the liver). Of the GLD genes, 22/60 had an eQTL. Next, we sought to understand the genetic architecture of the GLD module and link this architecture to the extensive database of risk loci identified for metabolic disease traits across many genome-wide association

and genome sequencing studies. To do so, we first constructed genetic models of gene expression to map eQTLs using MetaXcan[37] for all genes in the GLD module. We then explored the SNPs composing the genetic models of gene expression MetaXcan produced for association to human disease traits by incorporating GWAS summary statistics for the following traits of interest: LDL[22], HDL[22], total cholesterol[22], Triglyceride[22], HbA1c[23], and blood glucose[38]. These analyses helped establish whether variations in DNA that drive changes in gene expression also drive susceptibility to disease. We ran these analyses on the STARNET liver GLD genes, and find eight unique genes (DHCR7, FADS1, FADS2, FLVCR1, LSS, MMAB, MVK, and VPS37D) to be significantly and causally associated with LDL (FADS1, FADS2, and LSS), HDL (FADS1, FADS2, FLVCR1, MMAB, and MVK), total cholesterol (DHCR7, FADS1, FADS2, MMAB, and MVK), and TG (FADS1, FADS2, VPS37D; FDR < 5%; Supplementary Data 9). On average these eight genes explain 3.63% of the traits' variation, with LSS standing out as an outlier, explaining 26% of the variance of LDL cholesterol (mean variance explained without LSS is 2.39%). We further examined these genes with respect to HbA1c levels and found FLVCR1, FADS1, and FADS2 were significantly causally associated (FDR < 5%), although each explained <1% of the variation in HbA1c levels. While these results support causal associations between GLD genes and LDL, HDL, total cholesterol, and HbA1c, the approach assesses only a single gene at a time, whereas we identified these genes in a coherent regulatory network. To model the relationship between these genes and the common regulatory control that defines this network as a coherent unit, we constructed predictive causal network models related to the GLD module genes in which we considered the eQTL and associated causal relationships as prior input into this reconstruction process.

**Constructing probabilistic causal network models of the GLD module.** The GLD module not only represents canonical lipid biosynthesis pathways and known drug targets for lipid control, but also contains genes whose functions with respect to lipid biosynthesis are either not fully characterized or are completely unknown. In addition, the association of this module to glucose levels suggests unknown interactions in this module that may link lipid control to glucose control, highlighting that our understanding of lipid biology as represented by canonical pathways is incomplete. To expand our understanding of the canonical lipid metabolism pathways and their associations to glucose metabolism, we constructed four different probabilistic causal networks on the STARNET data using the RIMBANET software package that we developed and that we have previously applied to a wide array of diseases to identify networks and their master regulators underlying these diseases[31–33,39–45]. The first network we constructed was based only on the GLD module genes to elucidate the regulatory structure among these genes (referred to as GLD network). We then constructed a multiscale, integrative network comprised of the GLD genes, metabolites, and clinical traits to further elucidate the connections among GLD genes, metabolites, and clinical features (referred to as multiscale GLD network). Our third network construction took as input a GLD-focused expanded set of genes to explore whether genes outside the GLD module may modulate its state (referred to as expanded GLD network). And finally, we constructed a network based on all genes represented in the liver coexpression networks in order to place the GLD network and its regulators in a broader framework (referred to as global liver network).

In the GLD network, LSS, lanosterol synthase, is the most upstream gene (Fig. 4A). LSS catalyzes the cyclization of oxidosqualene into lanosterol, a key intermediate in cholesterol

biosynthesis. This gene has previously been identified as an important driver of various disease-related processes in liver as well as in other tissues[46], but it has not previously been associated with glucose metabolism. The multiscale GLD network (Supplementary Fig. 6A) is comprised of a single connected network component in which the GLD gene expression traits are linked to metabolites and clinical features. Interestingly, in the multiscale network the metabolites valine, leucine, and isoleucine are prominently positioned. These branched-chain amino acids directly connect the glucose (as measured by metabolomics) to the HbA1c clinical trait that in turn is linked to the GLD module genes. Importantly, branched-chain amino acid metabolism has previously been causally linked to the etiology of T2D[47].

To explore the molecular link between lipid and glucose metabolism, we expanded the GLD network by including 1592 genes across all other modules whose expression correlated with the first PC of the GLD module at an FDR < 5%. This gene list was then further expanded using the PEXA algorithm[48], a method for expanding gene lists by incorporating known interactions from KEGG and PPI networks. This expanded the GLD module to 3163 genes (Fig. 4B). To construct the global liver network, we took all the coexpressed genes in the liver (7646 genes) and expanded each module using PEXA, to realize the final expanded set of 8812 genes (Supplementary Fig. 6B and Supplementary Data 10).

In order to elucidate the regulatory framework of the GLD networks, we identified master regulators (key driver genes, or KDGs) predicted to modulate the state of the GLD genes across all networks we constructed, using an artificial intelligence key driver analysis (KDA) algorithm. We applied KDA to the GLD and multiscale GLD networks to identify master regulators within these networks, and then applied KDA to the expanded GLD and global liver networks to identify master regulators both inside and outside of the GLD network that modulated the state of the GLD network. KDA on the GLD network resulted in the identification of 4 KDGs (LSS, DHCR7, HMGSC1, and CYP51A1); in the multiscale GLD network three genes were found as KDGs (LSS, SNAI3-AS1, and HMGCS1). In the expanded GLD network 10 genes were identified as KDGs (all belonging to the GLD module), and in the global liver network, 26 genes were identified as KDGs of the GLD network (of which 9 belong to the GLD module). Of KDGs identified in the different networks, only two genes were overlapping across all four networks: LSS and HMGCS1 (Supplementary Data 11). To validate and prioritize the KDGs identified across all liver networks constructed in STARNET, a composite score reflecting the number of times a KDG was found in STARNET networks was computed (see Methods section; Table 1). Using this ranking system, LSS was identified as the most significant key driver gene in the liver and selected for experimental validation.

**Exploring the molecular connection between lipid and glucose metabolic co-regulation.** To characterize how LSS regulates processes associated with glucose metabolism, we examined the impact of LSS on the expanded GLD and global liver networks (Fig. 4B). In the expanded GLD network, the module that was most highly correlated with the GLD module was the bisque module (r value for pairwise correlation of PCs: 0.40, paired Student's t test FDR: 4.42E-20, two-tailed). When edges were collapsed at the module level, the GLD module had the most out edges to the bisque module (Supplementary Table 4). The bisque module contains many genes related to glucose and pyruvate metabolism including GCK, PKLR, KHK, and GPD1. Of these, PKLR, encoding the liver type pyruvate kinase, is of particular interest, as it is not only a key enzyme in glycolysis, but also a regulator of gluconeogenesis given its key role in pyruvate

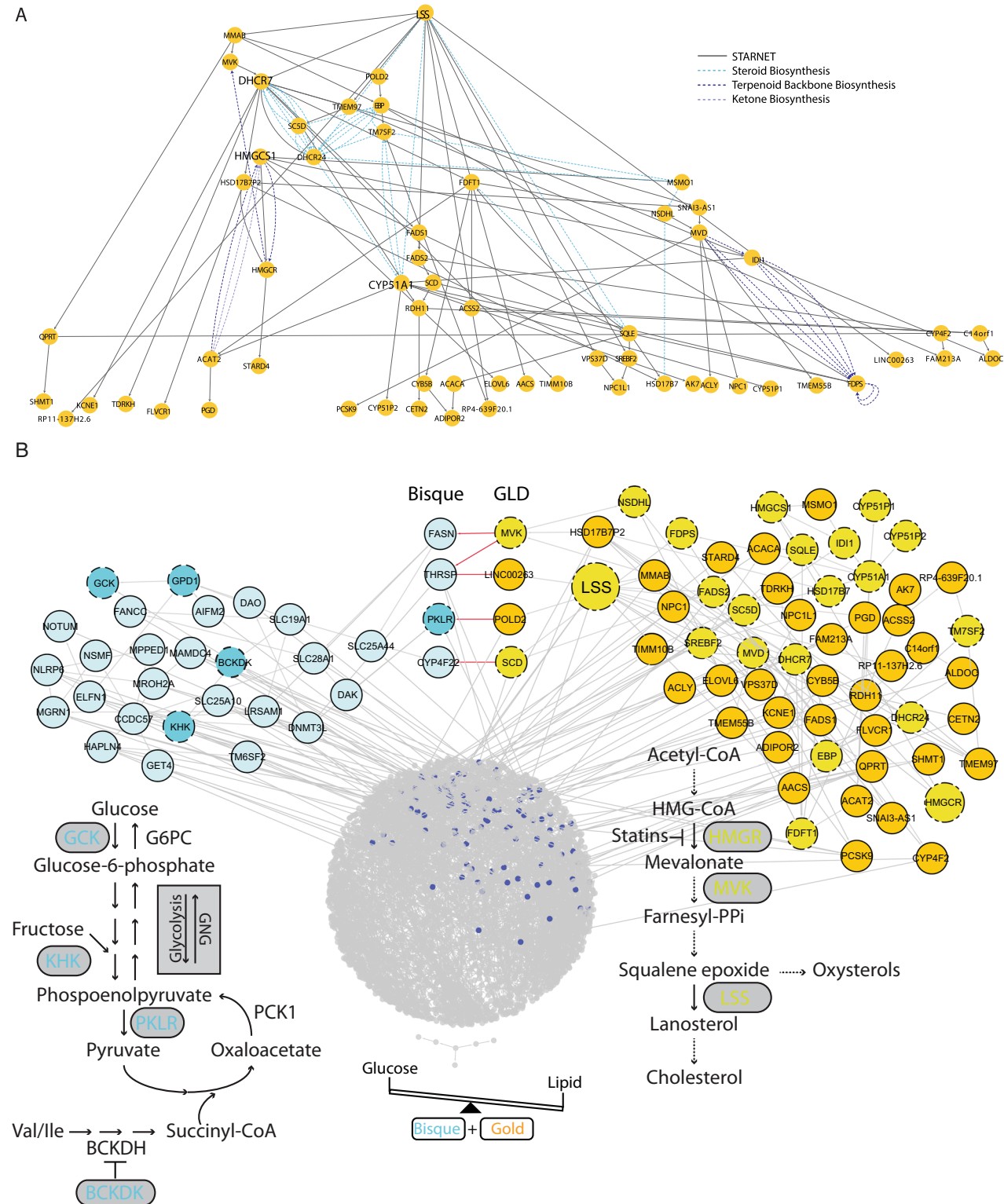

**Fig. 4 Probabilistic causal networks. A** Probabilistic causal network in the STARNET data for the GLD module genes only. Layout is in a hierarchical manner. As such, the nodes further towards the top are the most 'upstream'. Bold genes are the key drivers of this network. Dashed lines represent known edges from the KEGG pathways. **B** Expanded GLD-focused BN looking at the connections between GLD and bisque module (colored blue).

recycling[18] (Fig. 4B). We identified the bisque module contains the branched-chain ketoacid dehydrogenase kinase (*BCKDK*) gene. *BCKDK* regulates the branched-chain ketoacid dehydrogenase complex and therefore catabolism of branched-chain amino acids (BCAAs), the products of which eventually lead to the formation of substrates for lipogenesis, ketogenesis, or

gluconeogenesis in the liver. Importantly, key driver detection in the global BN revealed that *LSS* is predicted to be a KD of the bisque module as well as the GLD module (upstream of: *FASN, THRSP, PKLR, SLC28A1, FANCC, CYP4F22*, and *SLC25A44*; Fig. 4B). Thus, our network models support LSS as a master regulator of networks underlying lipid and glucose metabolism.

**Table 1 Top 15 Key Driver Genes based on a weighted score.**

| Gene | STARNET score |
| --- | --- |
| LSS | 3.9 |
| DHCR7 | 2.71 |
| HMGCS1 | 2.44 |
| IDI1 | 1.25 |
| TMEM97 | 1.24 |
| SC5D | 1.12 |
| ACACB | 0.92 |
| MVK | 0.88 |
| MMAB | 0.85 |
| CYP51A1 | 0.83 |
| MRPS15 | 0.77 |
| PTPRJ | 0.73 |
| ABCC6 | 0.69 |
| SNAI3-AS1 | 0.67 |
| SPG7 | 0.5 |

**In vivo drug perturbation of lanosterol synthase**. To test the effects of *LSS* as a GLD module key driver involved in regulating both cholesterol levels and glucose metabolism, we inhibited the enzyme in B6 mice and observed clinical and transcriptional effects. Our model predicted that inhibiting *Lss* would reduce the activity of the GLD module, resulting in decreased production of LDL cholesterol and increased glucose levels. Pharmacological inhibition of *Lss* was achieved by using BIBB-515, a compound that selectively inhibits Lss enzymatic activity. BIBB-515 was developed as a non-statin cholesterol reducing medication and has been shown to significantly disrupt hepatic cholesterol synthesis in rats and mice and rapidly decrease LDL cholesterol levels in hamsters. These earlier findings reinforce our model's prediction that inhibiting Lss lowers LDL cholesterol levels. In contrast, effects of Lss inhibition on hepatic gene expression and associated blood glucose levels have never been investigated. Our model predicted that inhibiting Lss should significantly alter expression of downstream GLD module genes, accompanied with increased levels of blood glucose. To validate these model predictions, we treated C57BL/6J (B6) mice with a diet containing BIBB-515 (55 mg/kg) for 10 days, and compared the plasma glucose and hepatic gene expression profiles to mice fed a control diet ($n = 11$ per group). Differential expression analysis showed a total of 1714 genes whose expression was significantly altered in mice treated with BIBB-515 (FDR < 0.05). Of these, 14 were orthologs of GLD genes, a substantially larger overlap than would be expected by chance (odds ratio: 2.46, Fisher's exact *p*-value 0.0059, one-tailed). Targeted measurement of fold-change in gene expression by quantitative PCR (qPCR) showed increased expression of GLD module genes involved in cholesterol synthesis in mice treated with BIBB-515: *Lss* (FC: + 1.9, Student's *t* test *P*: 0.004, two-tailed), *Dhcr7* (FC: + 1.9, Student's *t* test *P*: 0.0009, two-tailed), *Idi1* (FC: + 1.7, Student's *t* test *P*: 0.04, two-tailed), and *Hmgcr* (FC: + 2.4, Student's *t* test *P*: 0.0005, two-tailed; Fig. 5A). Other GLD module genes that also showed increased expression in response to Lss inhibition were *Pcsk9*, *Mvd*, and *Rdh11* (Supplementary Fig. 7). *Adipor2* was the only GLD module gene analyzed by qPCR that did not show a significant change in expression (Student's *t* test *P*: 0.06, two-tailed). In agreement with our STARNET models demonstrating correlation between expression of GLD module genes blood glucose levels, blood glucose level increased 14.5% in response to Lss inhibition by BIBB-515 (Student's *t* test *P*: 0.0006, two-tailed; Fig. 5B). Correspondingly, the expression of key gluconeogenesis genes, phosphoenolpyruvate (*Pck1*), and glucose-6-phosphatase (catalytic subunit *G6pc*), increased two- to three-fold in response to the Lss

inhibition (FC: + 3.0 and 2.1; Student's *t* test *P*: 2E-7 and 0.0006, two-tailed, respectively; Fig. 5C). In contrast, we observed no significant effects on levels of non-HDL plasma cholesterol in response to the Lss inhibition (Student's *t* test $p = 0.17$, two-tailed). This was not entirely unexpected given previously reports showing minimal response to lipid-lowering drugs in wild-type normolipemic mice[49]. The lack of response to lipid-lowering drugs in mice may have several explanations, including hepatic synthesis and assembly of apoB48-containing lipoproteins (apoB100 in humans)[50] and having HDL as its main carrier of blood cholesterol (LDL in humans). On this note, the Lss inhibition did increase the levels of HDL cholesterol (FC: + 10.0%, Student's *t* test *P*: 0.003, two-tailed) and in parallel total plasma cholesterol (FC: + 9.4%, Student's *t* test *P*: 0.014, two-tailed; Supplementary Fig. 8). Since the levels of HDL cholesterol is normally negatively correlated with LDL cholesterol, the increase in plasma HDL represents a perturbation of cholesterol metabolism in the predicted direction. Overall, the results from the experimental validation is consistent with our model's predictions that inhibiting Lss should perturb the GLD module genes and increase blood glucose levels. In addition, taking previous reports establishing that inhibition of Lss disrupts hepatic production of LDL cholesterol particularly in other species than mice and the reciprocal increase in HDL levels observed in our experiments together, the model prediction of decreased LDL was at least partly fulfilled.

**Discussion**

Through our integrative analysis of multiomic data generated on the STARNET cohort, we uncovered novel gene interactions and predicted new regulators of the complex and paradoxical links between T2D and LDL levels that impact CAD risk. Using the STARNET dataset, we demonstrated that of all the tissues profiled, liver harbored the most genes associated with lipid and glucose-related traits. By organizing the gene expression traits into coexpression networks, we discovered a module of genes coexpressed in liver – the GLD module – with an inverse relationship to plasma lipids and blood glucose levels. The GLD module is highly enriched for genes involved in cholesterol biosynthesis and contains the majority of the genes that encode for enzymes involved in the classic mevalonate pathway as well as its transcriptional regulator *SREBF2*. GLD also contains several genes involved in intracellular cholesterol transport, including *PCSK9*, *NPC1*, and *NPC1L1*. We speculate that the tight network of these genes specifically in liver is related to the crucial role that this organ plays in cholesterol and glucose homeostasis. Indeed, the liver does not only play a role in lipoprotein metabolism, but it also converts a significant amount of cholesterol into bile acids. Furthermore, these GLD genes are affected by lipid-lowering drugs and replicated in liver data from another human cohort and experimental mouse populations.

Given that the GLD module reflects interactions driven by a common regulatory framework, we employed probabilistic causal reasoning to construct causal network models of this module and the context in which it occurs in liver. Such probabilistic causal network models can elucidate the regulatory architecture of core subnetworks, leading to the identification of liver key driver genes (KDGs) predicted to modulate network states. Given the inverse relationship between LDL and HbA1c, we predicted that the KDGs of the GLD module would affect cholesterol and glucose metabolism through the GLD module genes and the larger network context in which the GLD subnetwork operates. In our liver networks, *LSS*, lanosterol synthase, is the most upstream KDG in GLD and the larger networks containing GLD (Fig. 4A), producing the highest KDG score out of the all KDGs in the network (Table 1).

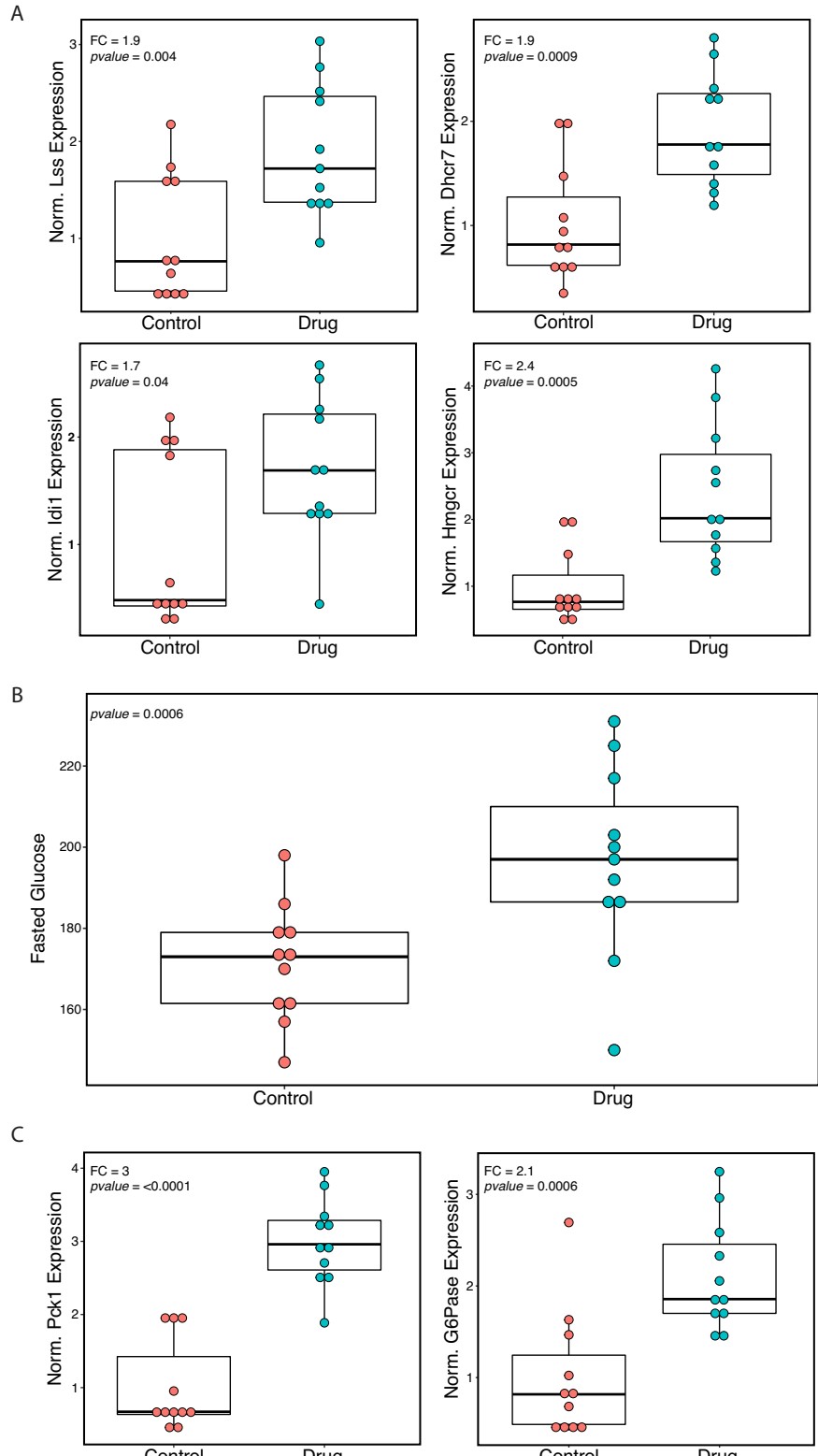

**Fig. 5 Inhibition of Lss in mouse model. A** Boxplots of qPCR expression for Lss, Dhcr7, Idi1, and Hmgcr measured from liver samples of B6 mice fed either chow (control) diet or drug (BIBB515, Lss inhibitor) diet. **B** Boxplots of fasted glucose measured from blood at the end of 10 days of diet. **C** Boxplots of two key gluconeogenesis genes, PEP Carboxykinase (Pck1), and Glucose-6-Phosphatase (G6Pase), measured from qPCR of liver samples. $n = 11$ animals with control diet and 11 animals with drug diet. Center line, median; box limits, upper and lower quartiles; whiskers, 1.5x interquartile range; points, all individual data points.

LSS catalyzes a crucial step in the biosynthesis of cholesterol and its downstream products include vitamin D, bile acids, oxysterols and steroid hormones. Lanosterol is the first cyclic sterol intermediate in the cholesterol synthesis pathway and is involved in post-transcriptional regulation of *HMGCR*[51]. *LSS* has been studied as a possible drug target for cholesterol lowering[52], and potent *LSS* inhibitors have been developed. Indeed, pharmacologic inhibition of *LSS* with BIBB-515 rapidly reduces LDL cholesterol levels in normolipemic hamsters[53]. In our mouse study, given liver *LSS* is a KDG of GLD, our model indicated that inhibition of liver *LSS* would alter the GLD network state, which in turn would lower cholesterol levels and increase glucose levels. We found that inhibition of hepatic Lss in our mouse model significantly perturbed the GLD network and increased glucose levels, as our model predicted. While total and LDL cholesterol were not reduced, as we would predict in humans, this may have been expected here since the metabolism and regulation of plasma cholesterol in mice is quite different from that in humans. Most notably, wild-type mice carry a majority of their plasma cholesterol in HDL particles, not in LDL particles as in humans. As a consequence, mice have low and stable levels of LDL cholesterol unless they have been genetically engineered to model hyperlipidemias. In particular, it has been shown that statins do not alter LDL levels in normolipemic mice[49]. Since our model predicts that the target of statins, *HMGCR*, is downstream from *LSS*, a similar unresponsiveness to inhibition of *LSS* could be expected. Nevertheless, it is clear that inhibition of *LSS* caused a significant perturbation in cholesterol synthesis. This is consistent with previously published results that inhibition of *LSS* measurably reduced the rate of LDL cholesterol synthesis in rats and mice, even though no change in the total levels of serum LDL cholesterol was reported[53].

While the effect of *LSS* on cholesterol metabolism has been well established, its effect on glucose metabolism has not been well studied. We identified that pharmacologic inhibition of *LSS* in mice caused increased levels of fasting blood glucose, paralleled by increased expression of two key gluconeogenic genes, *Pck1* and *G6pc*. These observations raise the possibility that GLD impacts blood glucose levels in part through *LSS* activity by changing the rate of gluconeogenesis. This is further evidenced by additional network analyses that identified the bisque module in liver as the most closely correlated with GLD. The bisque module contains key enzymes in glucose and pyruvate metabolism including pyruvate kinase. Although pyruvate kinase is classically viewed as a glycolysis enzyme, it is well established that pyruvate recycling via pyruvate carboxylase, phospoenolpyruvate carboxykinase (*Pck1*), and pyruvate kinase is an important regulator of gluconeogenesis under non-fasting conditions[18]. Bisque also contains the gene encoding glucokinase, which catalyzes the first step in glycolysis. Again, production of glucose through glucose-6-phospatase and hepatic reuptake is in part mediated through glucokinase (glucose cycling) and is an important determinant of gluconeogenesis. Thus, the opposing effects on glycolysis and gluconeogenesis are in part controlled by GLD, and bisque may ultimately determine net glucose production by the liver with subsequent consequences for levels of blood glucose (Fig. 4B).

In addition, the bisque module contains genes involved in branched-chain amino acid metabolism, which has been demonstrated to play a causal role in T2D[47]. Of note, the bisque module contains *BCKDK*, which encodes a kinase that inactivates the branched-chain ketoacid dehydrogenase complex, a key enzyme in branched-chain amino acid catabolism. Branched-chain amino acids are relatively abundant and their breakdown yields carbon that can be used for gluconeogenesis (valine and isoleucine, Fig. 4B). We speculate that the fine-tuning of carbon fluxes within the liver plays a role in the relationship between cholesterol and glucose metabolism.

Not only does the GLD module contain the target of statins, *HMGCR*[54], but also the targets of other lipid-lowering drugs such as *NPC1L1*[54] (ezetimibe), *ACLY*[55], and *PCSK9*[54]. The coherence of the GLD network module even when splitting the STARNET cohort by drug usage and HbA1c levels supports the importance of this network module to sustain proper balance between cholesterol and glucose metabolism in the liver. Furthermore, this establishes that the observed inverse relationships of GLD with lipid and glucose traits are not driven by the medication usage, even though medications such as statins can alter GLD through its impacts on HMGCR, which in turn may explain impacts on T2D risk.

In summary, using a data-driven approach we identify a small gene network that inversely regulates hepatic lipid and glucose metabolism. Our study highlights the capability of this multiscale, integrative network approach to uncover novel gene interactions, even in highly studied canonical pathways, that underlie the dual co-regulation of plasma lipids and blood glucose metabolism.

## Methods

**Normalization and QC**. For each tissue, the counts matrix were taken as the starting point (see Methods section from Franzen et al.[25]). From that, transcripts were filtered using the criteria of having at least 1 count per million in at least 10% of the samples in that tissue. After removing lowly expressed transcripts, counts were normalized using the Trimmed M Means methods using limma in R version 3.1.0. The normalized counts were then transformed to fit to a negative binomial distribution using the voom method from the same limma package[56]. Using variance partition, we found large batch effects in each tissue mostly due to the flow cell that it was sequenced on[57]. As such, for all tissues, we corrected for flow cell. For the multiple samples which were sequenced twice to increase coverage we made sure that it was counted as a part of both FC_1 and FC_2 instead of a third unique group "FC1_FC2". We further checked that all of these flow cell covariates were not co-linear, and if they were, we removed them. We also ensured that each group (flow cell) had at least 10 samples as we ran a linear model to correct for batch effect. After doing this, we ran PCA on the residuals (corrected for flow cell) to detect outliers; we defined outliers as samples, which were three standard deviations outside of PC1 and PC2. After removing these individuals, we re-corrected the data for only the remaining individuals again ensuring that there were at least five individuals in each flow cell and that none of the covariates were co-linear.

To make sure this was the best method for normalization, we ran differential expression analysis on the residuals against the different batch effects that we had (sequencing lab, protocol used, and hardware identifier). For two tissues, Blood and VAF, we found that there were multiple genes that were still significantly differentially expressed in regard to batch effect. Thus, in these two tissues, we re-corrected them adding in the covariate hardware-identifier.

**Differential expression**. We used the edgeR package in R version 3.1.0. In each tissue, we ran differential expression analysis on the normalized data against the clinical phenotypes of interest. *P*-values were adjusted for multiple testing using the BH method (FDR). For the differential metabolite analysis, the metabolites were first z-normalized.

**Enrichment analysis**. We annotated each gene with its associated GO categories and KEGG pathways using the goseq package in R version 3.1.0. We performed the enrichment analysis using the topGO package in R version 3.1.0, which performs a Fisher's Exact Test on each annotation category using the provided foreground and background gene lists and uses the Benjamini–Hochberg procedure to select significantly enriched annotations. We used as our foreground the list of significant trait-correlated DE genes in each tissue, and as our background the list of all genes expressed in the same tissue. Visualizations of GO terms were made using Revigo[58].

**Coexpression analysis**. We used the parallelized version of WGCNA in R version 3.1.1. (https://bitbucket.org/multiscale/coexpp). All parameters were default and we used "tree cut" algorithm to define clusters.

**Coexpression analysis for GTEx**. We collected gene expression data from the GTEx portal (Version 7)[59], and divided all individuals into two groups according to their chronological ages: (1) young group (age ≤ 35) and (2) old group (age ≥ 65). We chose to perform coexpression analysis only on tissues with at least 30 samples, as this is the approximate minimum number of samples required to compute a reliable correlation coefficient between expression levels[60]. There are 11 tissues with sample sizes in both young and old group ≥30, namely esophagus,

transformed fibroblast, lung, aorta, thyroid, tibial nerve, tibial artery, sun exposed skin, skeletal muscle, subcutaneous adipose, and whole blood, which were kept for further analysis. For the liver tissue, we split by median age of 55 for the young (≤55) and the old (>55) in order to have sample size greater than or equal to 30.

Similar to the GTEx study[59], for each tissue we only kept genes having at least 0.1 FPKM in 2 or more individuals and then normalized the expression of each gene (across samples) into a standard normal distribution. We adjusted for the following confounding factors: (1) gender, (2) collection center, (3) RIN, (4) ischemic time, and (5) three genotyping principal components. The genotype PCs were constructed using GCTA[61] by a few quality controls using plink (e.g., − MAF 0.1, − geno 0.05, −hwe 1e-6, −Chr 1–22). Finally, we constructed young and old networks for each tissue by weighted gene correlation network analysis (WGCNA)[29] with the soft powers to be 5 for all tissues.

**MetaXcan analysis**. We applied MetaXcan[37,62] to integrate GWAS summary level data and sample level STARNET eQTL data (genotype and gene expression) to identify genes underlying LDL[22], HDL[22], total cholesterol[22], triglyceride[22], HbA1c[23], and blood glucose[38] traits. Based on MetaXcan pvalue, false discovery rate was derived using Benjamini–Hochberg procedure.

**eQTLs**. Using fastQTL we called cis eQTLs for the tissue of interest[63]. First we checked the ancestry of our population and identified four individuals as outliers (non European). We then ran PEER to identify surrogate variables. Using the known covariates for batch effect as well as age, gender, and surrogate variable (SV) 1 through 20 we ran fastQTL. We ended up using 7 SVs for the LIV as that was where the number of cis eQTLs plateaued.

**Bayesian network**. RIMBAnet was employed to construct the probabilistic causal networks[39]. The input for this was the continuous gene expression values for every individual, as well as the same values discretized into three states. This enables the compute time to be tractable and reduces the search space for every gene to be defined as either high, no, or low expressed in each individual. We also used cis eQTLs as priors. Due to the central dogma of biology, we know that DNA ->RNA so the eQTLs can be used to break equivalent structures. In all, 1000 MCMC reconstructions were run and a posterior probability of 0.3 was taken as a cutoff for returning results. These parameters are based on a previously reported simulation study, where using a posterior probability cutoff of 0.3 in 1000 reconstructions was found to provide the best balance between precision and recall[40,41]. After the 1000 reconstructions, the networks were all merged together, and cycles identified and broken by removing the "weakest link" (edge with the lowest posterior probability). This results in a network that is a directed acyclic graph (DAG) and a Bayesian Network.

**Discretization**. Every gene was discretized into three states of expression: high, none, or low. This was done using Matlab and k-means clustering with a $k = 3$. First, each gene's expression was normalized to a normal distribution and then clustering was run to find the three clusters. If there were only two clusters detected, we allowed two (high, low) and did not force a third cluster.

**Causal inference test**. In order to identify if there were any prior edges of SNP ->gene -> metabolite, or SNP ->metabolite -> gene, we ran the causal inference test (CIT)[64,65]. For the genes of interest, we identified the top eSNP (SNP that had the highest eQTL for that gene). We then took those corresponding SNPs and looked to see if any were associated with the metabolites (FDR < 5%). Further, we took the metabolites and the genes and checked their correlation. This resulted in 3,410 trios of SNP - gene - metabolite trio that were all correlated. For each of these, we ran the CIT test in R to identify both the causal model (SNP ->gene -> metabolite) and the reactive model (SNP -> gene -> metabolite). If the causal omnibus $p$-value (p_cit) that was significant after multiple testing and the reactive omnibus $p$-value was not significant, we called that edge a prior edge for gene -> metabolite. And, if the reactive omnibus $p$-value was significant and the causal omnibus $p$-value was not significant, we called the edge of metabolite -> gene significant[25]. We identified 95 causal interactions of SNP -> gene -> metabolite.

**Multiscale network**. For the multiscale network, we used RIMBAnet as well. The data was discretized in the same manner as above. For clinical traits, the same process was followed and categorical traits were assigned to 2 states (yes and no) or 3 if there was a third option such as "quit" or "maybe". Priors from the CIT test were inputted as "strong priors". These edges were forced to be present in each of the 1000 independent networks in the seed network with a probability of 1. All 95 edges inputted as prior edges remained in the consensus network.

**PEXA analysis**. We applied the PEXA algorithm[48] to select genes to include in our expanded networks. PEXA takes a seed list of genes, which in the original PEXA publication are genes identified by an siRNA screen, and uses KEGG pathway data and protein–protein interaction data to select related genes. For the GLD expanded network, our seed list contained all genes in the GLD module, and all genes whose expression in LIV was correlated with the first PC of the GLD

module at an FDR < 5%. For the global liver network, we ran PEXA separately for each of the 28 modules in LIV identified by WGCNA, including the GLD module. For each run, we used as our seed list all the genes in that single module. We then combined the resulting 28 expanded gene lists to produce a single global expanded gene list.

**Key driver analysis**. We performed key driver analysis (KDA) to detect key drivers of networks of interest using the KDA software package[30,31,66]. In the gene only networks, we identified key drivers of the GLD module genes at a maximal path of 7. For multiscale network, we ran KDA to identify KDGs of the GLD genes, KDGs of the clinical traits, and KDGs of the metabolites.

**Key driver ranking**. For every network, we took the KDGs and their ranking based on $p$-value (or number of targets downstream if the $p$-value was 0) with the least significant being at top and then divided the rank by the total number of KDGs to get a weighted KDG. This method allowed for the top KDG to have a rank = 1. We then summed the ranks across the four BNs for the STARNET data to come up with a score for each gene with the max possible being 4 (would have had to been the #1 KDG in all 4 BNs).

**Animal studies**. Six-week-old female C57BL/6J (B6) mice were obtained from The Jackson Laboratory (Bar Harbor, ME) and acclimated for 2 weeks prior to initiation of experiments. Mice were housed three or four per cage and maintained at 22 °C under a 12-h light cycle. Mice were maintained in accordance with University of California, Los Angeles Institution Animal Care and Use Committee protocols. Three days prior to initiation of experiments, all mice were switched from a chow diet (Ralston Purina Company) to the control diet (Research Diets D111112201; kcal = 15% fat, 65% carbohydrate, 20% protein). At experiment initiation, mice (n = 11/group) were randomized to control diet or a matched diet containing BIBB-515 (dosed at 55 mg/kg body weight).

**Hepatic RNA isolation and gene expression analysis**. Flash frozen hepatic samples were homogenized in Qiazol (QIAGEN) and RNA isolated according to the manufacturer's protocol with RNeasy columns (QIAGEN). Isolated RNA was converted to cDNA using a High Capacity cDNA Synthesis Kit (ThermoFisher) according to the manufacturer's instructions. Primers targeted at genes of interest were designed and reactions were run in triplicate using KAPA SYBR FAST qPCR Master Mix (Roche) with 8 ng of cDNA per reaction. Quantification was performed on the LightCycler 480 (Roche) with an initial 95 °C step followed by 45 cycles of denaturation at 95 °C for 45 s, annealing at 57 °C for 45 s, and extension at 72 °C for 45 s. Melting curves were performed by 95 °C for 5 s, followed by 65 °C for 60 s, and a continuous read step of seven acquisitions per second between 65 °C and 97 °C. Results were expressed as averages of three independent reactions and normalized to *B2M* as a housekeeping gene.

**Blood glucose and plasma cholesterol**. Retro-orbital blood was collected used isoflurane anesthesia after 5 h of fasting. Blood glucose was determined immediately following blood collection using a AlphaTrak glucometer (Zoetis). Total and HDL plasma cholesterol were determined by enzymatic processes employing colorimetric end points[67,68]. Samples were diluted with saline so their measured absorbance values were within the linear range of established standard curves. Measured absorbances were divided by the absorbance of a single standard concentration near the midpoint of the linear range of the established standard curve. The product of this factor and the standard concentration yielded the unknown value. Values were expressed as milligrams per deciliter. Each sample was measured in quadruplicate. An external control sample with a known analytic concentration.

(Accutrol, Sigma No. 2034) was run in each plate to assure accuracy. TC was determined according to Allain et al.[69], as provided in a kit (Sigma procedure 352). HDL-C was separated from LDL-C + VLDL-C by precipitating the latter with phosphotungstic acid and magnesium chloride according to Marz and Gross[70] and Assmann et al.[71]. Precipitation reagents were obtained in a kit (Sigma procedure 352-4). All reaction volumes were scaled down from recommended levels so that the entire reaction could be run in a 96-plate microtiter well.

**Reporting summary**. Further information on research design is available in the Nature Research Reporting Summary linked to this article.

## Data availability

Data from the STARNET study are available through the Database of Genotypes and Phenotypes (dbGaP) under accession phs001203.v1.p1. Data from the mouse study are available through the Gene Expression Omnibus (GEO) under accession GSE157223. Data from the Molecular Signatures Database (MSigDB) are available at https://www.gsea-msigdb.org/gsea/msigdb. Data from GTEx are available at https://gtexportal.org/. Data shown in Fig. 2a, c are included as Supplementary Data 1. Data shown in Fig. 2b are included as Supplementary Table 1. Data shown in Fig. 3a are included as Supplementary Data 7. Data shown in Fig. 3b are included as Supplementary Data 6. Data shown in Fig. 4 are included as Supplementary Data 10.

## Code availability

Code sharing not applicable to this article as the conclusions do not rely on any custom software. All software used in the analyses reported in this article has been previously published, and is either publicly available or is available on request from its original authors. Details on how these previously published software packages were used are provided in Supplementary Note 1.

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

## Acknowledgements

We thank all participants for contributing to this research. We acknowledge research support from NIH R01HL125863 (J.L.M.B.), HL030568 (J.L.M.B.), and GM115318 (A.J.L.), American Heart Association A14SFRN20840000 (J.L.M.B.), Swedish Research Council 2018-02529 (J.L.M.B.) and Heart Lung Foundation 20170265 (J.L.M.B.), Foundation Leducq PlaqueOmics: Novel Roles of Smooth Muscle and Other Matrix Producing Cells in Atherosclerotic Plaque Stability and Rupture, 18CVD02 (J.L.M.B.), Foundation Leducq CADgenomics: Understanding CAD Genes, 12CVD02 (J.L.M.B., A.J.L.) and Astra-Zeneca, Molndahl, Sweden (J.L.M.B.). This work was supported in part through the computational resources and staff expertize provided by Scientific Computing at the Icahn School of Medicine at Mount Sinai. STARNET study was approved by IRB 154/7(2006), 188-M12 (2009), 277/T17 (2018) by the Research Ethics Committee of the University of Tartu (UT REC).

## Author contributions

A.T.C., W.T.B., D.M.J., J.C.K., A.J.L., J.L.M.B., and E.E.S. conceived, designed, and managed the study. A.T.C. performed statistical and computational analyses under the direction of E.E.S., with advisory input from NBD. W.T.B. performed all biological experiments under the direction of A.J.L., with experimental support from J.L.M.B.; STARNET data was collected and processed by O.F., S.K., C.W., K.S., R.E., A.R., K.H., L.M.G., and C.G. under the supervision of J.L.M.B.. D.M.J., N.D.B., G.M.B., A.W.C., E.E.K., C.A.A., S.M.H., R.D., J.Z., and Z.T. provided computational support and technical assistance. GTEx analysis was performed by J.Y. under the supervision of CAA and Z.T. MetaXcan results were performed by K.H.; A.T.C., DMJ, J.L.M.B., and EES wrote the manuscript, with input from W.T.B., AJL, C.A.A., S.M.H., and C.G. All authors critically reviewed the manuscript and contributed significantly to the work presented in this paper.

## Competing interests

J.L.M.B. and AR are shareholders and part of the board of directors in Clinical Gene Networks AB (CGN). CGN has an invested interest in the STARNET database. The remaining authors declare no competing interests.
