## [Peer Review File · Nature Communications]

Reviewers' comments:

Reviewer #1 (Remarks to the Author):

This is a very interesting and timely study by leading scientists in the field. The STARNET cohort is excellent for these studies. The techniques and bioinformatics used are all excellent. Still, I have several comments:

1. Diabetic dyslipidemia is mainly characterized by hypertriglyceridemia, prolonged postprandial hyperlipidemia, small dense LDL and low HDL-C. The accumulation of sdLDL and the low HDL-C are secondary to the action of CETP and the accumulation of TLRs. The key driver seems to be hyperTG. However, triglycerides are not included in the analyses ("We then explored the SNPs composing the genetic models of gene expression MetaXcan produced for association to human disease traits by incorporating GWAS summary statistics for the following traits of interest: LDL, HDL, total cholesterol, HbA1c and blood glucose"). It's critical that triglycerides are included in the analyses.

2. "We divided the STARNET cohort into two groups: one comprised of individuals reported as taking a lipid lowering drug (N = 334) and the other comprised of those not taking such drugs (N = 182). There are (as mentioned in the Discussion) several classes of lipid lowering drugs. They function differently, and the analyses should be performed on the individual classes of lipid lowering drugs. This includes both metabolomics ("Turning to the metabolomics data revealed that 110 metabolites were differentially expressed between STARNET individuals currently taking cholesterol lowering medications versus those not on such medications (Supplementary Table 4)); and coexpression networks.

3. This reviewer is puzzled by the use of apoE-deficient mice. These mice are not relevant for diabetic dyslipidemia and for human pathophysiology. Indeed, "in contrast to GLD in STARNET, in the mouse the liver module was positively correlated with a variety of lipid traits. This difference may relate to the super-physiologic levels of plasma cholesterol in Apoe^{-/-} mice (1000 mg/dl versus 300 mg/dl as typically observed in humans with familial hypercholesterolemia), as the lack of Apoe in these mice results in near blockage of VLDL and LDL particle uptake primarily in the liver.". Despite this, the Apoe^{-/-} mouse intercross populations were used to validate and prioritize the KDGs. This weakens the study.

4. How specific is the LSS inhibitor?

5. Overall, the phenotypic description of the STARNET cohort is weak.

6. The authors state that "SNPs proximal to HMGCR, the enzyme targeted by statins, have been shown to decrease LDL and cause increased T2D risk." To my understanding, the studies have found associations, but no real causality has been proven.

Reviewer #3 (Remarks to the Author):

The authors elucidated the perplexing juxtaposition between LDL and T2D in CAD patients. They used systems biology tools for the integration of several publicly available datasets and identified a small subnetwork that might be a key module for regulation of both cholesterol and glucose metabolism. They also performed additional analysis using Apoe^{-/-} mice model and tried to validate their finding in a mouse experiment. However, the results of the analysis in mouse model is not convincing. Overall, the manuscript is very well written, and the story line is quite clear. However, the authors should put additional attention in the validation of their results.

I have number of comments for further improvement of the manuscript:

Major concerns:

1. The authors should present the changes in the plasma or tissue more carefully. It is not clear in the manuscript. Figure 1A is also misleading. I suggest the authors clearly present the

tissue/plasma transcriptomics and metabolomics data is used. It is so much confusing in the paper.

2. This study employed a lot of computational analysis done by a series of packages and the key results are heavily dependent on these analyses. Therefore, it would be essential and appreciated if the author release the script for all the computational analysis so that the readers could generate the results presented in the paper.

3. The module identified by mouse dataset is different from the ones from human cohort, and the author claimed that the reason might be 'the super-physiologic levels of plasma cholesterol in Apoe^{-/-} mice'. However, in their experimental validation, they fed the mice with chow diet but the correlation with plasma cholesterol level is still negative. This inconsistency needs to be properly discussed. The authors may even remove the analysis of Apoe^{-/-} mice model since it does not fit in the paper. I also suggest the use of liver transcriptomics data presented in Jake Lusis' Cell Systems paper for providing additional support for their findings.

4. The authors analysed gene expression data from 7 different tissues and reported LSS is the master regulator of the glucose and lipid metabolism based on liver tissue analysis. Similar analysis have been done in a manuscript by Lee et al, Mol.Syst.Bio and LSS have been reported as the regulator of lipid and glucose metabolism in almost all tissues analysed in the paper. The authors should be careful in reporting the role of LSS in all tissues since its role is not specific to liver.

5. The last part of the results section should be rewritten carefully. The model predictions and the experimental data should be presented clearly in Results section and discussed in the last part of the paper.

The key finding in this study is the GLD module and its positive correlation with cholesterol metabolism and negative correlation with glucose metabolism. But the experimental validation in mice failed to demonstrate that.

In addition, the authors claimed in line 335 that 'our model predicted that inhibiting LSS would down regulate pathways associated with lipid metabolism that would result in increased cholesterol and increased glucose levels', which is very confusing since the correlation to cholesterol was suggested to be positive.

The authors simply stating that the validation of the study is not performed properly. Please can you clarify this statement. Such statement is quite disappointing.

"We found that LSS in our mouse model significantly perturb the GLD network as our model predicts, and that while glucose levels were increased as predicted, total cholesterol was not reduced as expected. That cholesterol levels were not reduced may largely reflect the inadequacy of mouse as a model of cholesterol control in humans".

6. There is also a fundamental issue in the analysis performed in the paper. I do not think an intervention increasing the plasma glucose can provide efficient treatment strategy. The authors should discuss this in the paper. In my opinion, inhibition of LSS in all tissues should not be presented as a good strategy. My question to the authors is if they would run a clinical study using LSS inhibition?

Minor concerns:

1. It might be very obvious to the authors, but 'OR' need to be annotated as 'Odds Ratio' when first time used.
2. In line 167, the author mentioned the GTEx cohort is separated into two groups based on age, which is very strange to me and need justification. The relationship between age and the biological

question in this study is not sufficiently discussed.

3. In line 208, I would suggest to use 'significantly changed' instead of 'differentially expressed' since metabolites are not expressed by the cells.

Reviewer #4 (Remarks to the Author):

This is an interesting manuscript, which leverages integrative approaches in STARNET (and other) cohort to identify a liver gene co-expression network, and therein the LSS gene, underlying both hepatic lipid and glucose metabolism pathways.

I believe the manuscript can be improved by (1) explaining the integrative analyses, (2) making all results/data available to the reader, and (3) performing additional experiments to corroborate the Authors' key conclusions. In its current form the manuscript is difficult to follow, even for a reader (as myself) who is familiar with the kind of integrative approaches and network-based analyses presented here. In my detailed comments (below), I highlight several areas for improvement, and I suggest ways to better the presentation of the results. I also believe that some methodological details and statements in the manuscript should be better justified, especially for a non specialised audience. The in vivo validation experiments can be also strengthened.

Detailed comments

The explanation of the data-driven computational strategy and integrative network approaches presented in Fig. 1 and in the text (page 5, first paragraph) can be improved by reporting (within the figure) the number of DE genes, networks (replicated and non-replicated), and KDG identified at each step of the analysis, and which crucial statistical thresholds were used at each step. This will help the reader to assess how the KDE are selected for in vivo validation starting from a transcriptome-wide analysis, highlight the support provided by integration of independent datasets, and outline the crucial steps in the KDE prioritization strategy.

The use of terminology DE signature and DE genes should be used cautiously to avoid confounding the reader. Typically, DE refers to a contrast between two states (e.g., case/controls) and the general reader will assume canonical differential expression analysis (or fold change analysis) between two conditions. Here "DE signatures" refer to genes whose expression correlates with a given trait. Therefore, sentences like "the largest DE signature, comprised of a total of 4,653 unique DE genes" can be confusing. To improve readability throughout the whole manuscript (and supplementary materials), I suggest to use "gene signature correlated" with trait or "DE signature correlated" with trait, and to specify "correlated with" all times this is required to clarify the nature of the DE gene signatures.

For each DE signature, the complete list of genes correlated with any trait in any tissue should be made available to the reader. In addition, a detailed summary of the overlap between the different DE signatures across tissues should be also made available. (Ideally, a small web portal to browse all DE signature genes and their correlations with traits will really help the reader to access and browse these extensive results.)

Related to the previous comment, in Figure 1 I assume the numbers in between the colors (e.g., in LIV, number 5 between 1243 and 1116) represent few DE signature genes that are too small to be evident in the figure. Even at high magnification, I cannot see colors (trait) associated with these numbers. Are the 5 DE genes in LIV (between 1243 and 1116) correlated with bl.glucose? The same question applies to the 2 DE genes in VAF.

"We ran gene ontology (GO) enrichment analysis for each DE signature in each tissue to assess

their biological relevance". Which method (tool) was used for this analysis? Most critically, what is the gene set background used in this enrichment analysis? Is the gene set background tissue specific? Does the gene set background account for the number of genes expressed in each tissue or across all tissues? These details are missing in the methods and are important to assess whether the GO enrichments are specific to given tissues.

It will help if the same trait nomenclature is used consistently between the main figure and Supplementary Table 1 (and elsewhere throughout the whole manuscript as well as in supplementary materials).

In the sentence "Genes in the glucose and lipid trait DE signatures were significantly enriched for 586 common GO terms, reflecting a significant enrichment of common GO terms (Fisher's exact test OR: 108.8 $P < 3E-324$), reflecting a high-degree of interaction between pathways associated with lipid and glucose biology", are the Authors referring to DE signatures across all tissues? This should be specified here. Beyond those related to glucose and lipid traits, are there other significant GO terms overlaps? A systematic analysis of all GO overlaps should be reported (supplementary). In addition, given that these 586 common GO terms are detected for the glucose and lipid trait DE signatures across all tissues, an appropriate background gene set used in the GO enrichment analysis should be the common set of genes expressed in all tissues. To reinforce this initial observation (Suppl. Fig. 1) the authors should test how the GO enrichment results are affected by the background gene set used in the analysis (tissue-specific vs across-tissue).

"We identified 20 modules across the liver (N= 6), VAF (N= 5) and SF (N=9) that were enriched for both lipid and glucose DE signatures...". The full list of WGCNA modules, the genes therein (including those that are part of the DE signatures), the complete GO enrichments for all modules and the genes contributing to the GO enrichment should be provided to the reader (Supplementary or via web portal).

How was the direction of the correlation between the modules and the trait been assessed? In Suppl. Table 2 and if possible in Fig. 2B too, it will help to have clear representation of the degree to which these modules are correlated (positively or negatively) with each trait. The correlation is reported in Fig. 2B (red-blue color scale), yet it appears that in most cases the R^2 are rather small. E.g., for the LIV gold, it seems that none of the correlations are greater than 0.3, which is R^2 less than 10%. Given that the 1st PC used to assess these correlations explains ~52% of variation and R^2 is less than 10%, are these correlations biologically meaningful? Also, what about the second (and third) PC? These PCs capture less variance of the module gene expression variation, but might show much higher correlations with the traits. Lastly, it will be also useful to have for each gene in the module, the raw gene-traits correlation statistics, in particular for the GLD module.

The correlations between the metabolites and PC 1 of GLD module are weak. From Suppl. Table 4, the maximum R^2 with any metabolite is ~10%, and given that PC 1 explain 57% of the variance of GLD module, these associations are very modest, and hence their biological significance is questionable. For instance, among the results highlighted in the text, the "positive correlation with Glucose" has $r=0.17$ (i.e., $R^2 = 3\%$) and it is not differentially expressed between STARNET individuals taking cholesterol lowering medications vs those not on such medications. Can these associations with metabolites be independently verified?

"We constructed coexpression networks for each tissue, identifying 40, 56 and 32 modules in liver, subcutaneous fat and visceral fat, respectively." Again, all data should be made available in full to the reader, including detailed statistics on the overlap with the GLD module. Related to this, are there other modules (in addition to GLD) that overlapped with the modules identified in 600 obese patients?

"...correlating the first principal component of this liver module with clinical metabolic traits

measured in these 600 individuals found positive correlations with glucose levels and a negative correlation with plasma LDL levels (FDR < 5%)". More details are also needed here, and in particular with respect to the magnitudes of these correlations. All data should be made available to the reader. Do the authors detect the same metabolite-module correlations as in Supplementary Table 4? A detailed metabolite-specific comparison should be carried out here.

"the 12 tissues currently represented in GTEx with sufficient sample sizes to construct coexpression networks". Why sample sizes greater than or equal to 30 is sufficient to construct coexpression networks? Is this sample size based on power calculation or other empirical evidences?

Since the Authors used "the STARNET cohort, given that CAD provides a more extreme context in which metabolic traits such as cholesterol levels are more significantly perturbed", and replicated their findings in another disease cohort (600 morbidly obese patients undergoing bariatric bypass surgery), what is the rationale to seek further replication in GTEx? Indeed, replication in tissues from healthy subjects (GTEx) plays against the initial Authors' rationale (with which I agree) to consider cohorts where cholesterol levels are more significantly perturbed.

"Across the 58 liver and 38 adipose modules identified, only a single module was identified as enriched for GLD module genes at an FDR <5%". All data should be made available to the reader. What is the overlap with the modules identified 600 morbidly obese patients and overlapping with GLD? A detailed three-way comparison would be useful here. Are there any other modules (apart from GLD) that have significant overlap between STARNET, 600 morbidly obese patients and the Apoe-/- mice on a high fat diet?

"Consistent with STARNET, at an FDR <5%, the first principal component of this mouse liver module was positively correlated with glucose and insulin traits." Again - more details and analyses are needed here. See also previous comments.

"The GLD module correlations to the lipid and glucose traits were also largely conserved across the different groups" and data presented in Fig. 3B; all specific correlations should be reported in detail. Again it appears these correlations are weak.

"Leveraging DNA information allows us to determine causality, as we know that DNA must be upstream of gene expression". I found this phraseology a bit unspecific. What is the meaning of "DNA must be upstream of gene expression"?

Expression QTL analysis STARNET liver samples: I suppose this analysis is limited to cis-eQTLs; what is the cis-window used to call eQTLs? Does the choice of the cis-window affect the number of detected eQTLs? Which FDR correction was used here? Does this FDR correction account for LD structure between the SNPs in the cis-window? Why did the Authors completely disregard trans-eQTLs here? The important role of trans-eQTL as drivers of known disease associations (including cholesterol metabolism) have been previously demonstrated (e.g., PMID:24013639).

In addition, as the focus here is on GLD, a co-expression module which should indicate co-regulation, this can be investigated by mapping trans-acting regulators of the GLD module, as previously shown (e.g., PMID:21572415).

"To model the relationship between these genes and the common regulatory control driving the coherence of this network,...". What do the Authors mean by "the coherence of this network"?

Bayesian network analysis - a posterior probability of 0.3 was taken as a cutoff for returning results from the MCMC reconstructions. This posterior probability threshold seems very low, and here the reader has to believe this threshold is appropriate without any detail supporting this specific choice. Can the Authors better justify this choice (also, in context of successive removal of

the “weakest link” (edge with the lowest posterior probability)?

“This gene list was then further expanded using the PEXA algorithm, to incorporate known interactions from KEGG and PPI networks to 3163 genes (Fig. 4B)”. First, the pathway expansion analysis (PEXA) in ref.48 was implemented to leverage KEGG pathway database and PPI data as orthogonally generated sources of information to identify genes involved in processes associated with insulin signaling based on siRNA screening data sets. How was the PEXA algorithm used here? (This is not described anywhere in the manuscript). Second, Figure 4B is not very clear and its legend needs to include more details. The larger network contains >3000 genes, and in the figure only the GLD and Bisque modules are extracted to link genes involved in cholesterol and glucose pathways? I believe that this kind of presentation of results can be misleading since at this point in the manuscript one can identify many more pathways linked through 3163 genes (not just the cholesterol and glucose pathways). How are these two pathways extracted from the large network of 3163 genes?

The data reported in Supplementary Fig. 5B are not very useful - what is the reader supposed to learn from the “hairball” network reported in panel B? This expansion of the GLD network to the global liver network does not seem to add much information here.

The list of identified KDGs is presented in Supplementary Table 7. In order for the reader to interpret and understand the data presented, it is important the Table's contents are clearly explained. Since not all readers might be familiar with the KDA analysis and its parameterization, each table heading (downstream signature_in_subnetwork, subnetwork_size, signature_in_network, network_size, signature, optimal_layer, etc.) must be adequately explained. In addition, a p-value = 0 (as for pvalue_whole, pvalue_subnet or pvalue_corrected_subnet) does not make sense, and this should be better replaced with the lower bound estimate of the p-value (e.g., it should be reported as “p<...”).

To validate and prioritize the KDGs identified across all networks constructed in STARNET, KDGs were predicted in the Apoe^{-/-} mouse data. What about the cohort of 600 morbidly obese patients (which was also used to replicate the GLD module)? It will be more powerful to seek replication of the KDGs in human cohorts (i.e., 600 morbidly obese patients) rather than in Apoe^{-/-} mouse alone. This might also help in better assessing the relative importance of DHCR7 vs LLS, as both these genes rank high and are predicted to be KDG in the Apoe^{-/-} mouse data (Table 1).

“...the module that was most highly correlated with the GLD module was the bisque module (r value: 0.40,...”. How was this correlation calculated? Also, what is the null hypothesis in this case? (is $H_0 r \neq 0$)? Since all these modules are within the expanded GLD network, what is the average (expected) correlation to be used to assess the significance of each module correlations?

Something is wrong with the legend of Supplementary Table 8. “Liver modules correlated to GLD module at FDR <5% and the number of edges to & from GLD genes in the expanded and global BNs. Type or paste caption here. Create a page break and paste in the Figure above the caption.” As mentioned earlier, and FDR = 0 does not make sense.

“Importantly, key driver detection in the global BN revealed that LSS is predicted to be a KD of the bisque module as well as the GLD module”. Maybe I am missing something here, but where are the results (statistics) of the KDG analysis for LSS in the bisque module reported? In Supplementary Table 7 (GlobalBN tab) only KDG results referring to the gold module are reported and in Fig. 4B there are no clear data indicating LSS is a KD in bisque module.

Given the suggested role of LSS as a master regulator of networks underlying both lipid and metabolisms, in order to assess the contribution of LSS to liver gene expression and demonstrate its master regulatory role of GLD module, the Authors should carry out a comprehensive transcriptomic analysis (e.g., by RNA-seq) in the liver (rather than looking at just 4 additional

genes, with 1(out of 4) gene showing no significant changes in expression). Besides, this will also allow to assess specificity and magnitude of the predicted effect upon in vivo drug perturbation of LSS, including the suggested role of LSS as a KD for the bisque module. I believe qPCR data on 4 GLD module genes are insufficient to state that "altering Lss perturbed the GLD module genes".

In addition, to further strengthen their conclusions, a dose dependent response analysis for both glucose and transcription in the liver would be beneficial here, although I understand this is an substantial experimental undertaking.

I hope my detailed comments and suggestions will help the Authors improving the manuscript, and importantly to make it more accessible to the wider readership of Nature Communications.

Reviewer:
Enrico Petretto, PhD
Duke-NUS Medical School (Singapore)
Imperial College London (UK)

This is a very interesting and timely study by leading scientists in the field. The STARNET cohort is excellent for these studies. The techniques and bioinformatics used are all excellent. Still, I have several comments:

1. Diabetic dyslipidemia is mainly characterized by hypertriglyceridemia, prolonged postprandial hyperlipidemia, small dense LDL and low HDL-C. The accumulation of sdLDL and the low HDL-C are secondary to the action of CETP and the accumulation of TLRs. The key driver seems to be hyperTG. However, triglycerides are not included in the analyses ("We then explored the SNPs composing the genetic models of gene expression MetaXcan produced for association to human disease traits by incorporating GWAS summary statistics for the following traits of interest: LDL, HDL, total cholesterol, HbA1c and blood glucose"). It's critical that triglycerides are included in the analyses.

Thank you for this comment and your positive reviews of our paper. We included TGs for all other analyses but thanks to your comment we have re-run the MetaXcan analysis to include TG as well. Please see the updated Supplemental Table 12 (formerly ST6), which includes the three genes we identified from the GLD module associated with TGs (*FADS1*, *FADS2* and *VPS37D*) and lines 260 – 271 in the text (copied below).

"We then explored the SNPs composing the genetic models of gene expression MetaXcan produced for association to human disease traits by incorporating GWAS summary statistics for the following traits of interest: LDL¹, HDL¹, total cholesterol¹, Triglyceride¹, HbA1c² and blood glucose³. These analyses helped establish whether variations in DNA that drive changes in gene expression also drive susceptibility to disease. We ran these analyses on the STARNET liver GLD genes, and find 8 unique genes (*DHCR7*, *FADS1*, *FADS2*, *FLVCRI*, *LSS*, *MMAB*, *MVK*, and *VPS37D*) to be significantly and causally associated with LDL (*FADS1*, *FADS2*, *LSS*), HDL (*FADS1*, *FADS2*, *FLVCRI*, *MMAB*, *MVK*), total cholesterol (*DHCR7*, *FADS1*, *FADS2*, *MMAB*, *MVK*), and TG (*FADS1*, *FADS2*, *VPS37D*) (FDR <5%; **Supplementary Table 6**). On average these 8 genes explain 3.63% of the traits' variation, with *LSS* standing out as an outlier, explaining 26% of the variance of LDL cholesterol (mean variance explained without *LSS* is 2.39%)."

2. "We divided the STARNET cohort into two groups: one comprised of individuals reported as taking a lipid lowering drug (N = 334) and the other comprised of those not taking such drugs (N = 182). There are (as mentioned in the Discussion) several classes of lipid lowering drugs. They function differently, and the analyses should be performed on the individual classes of lipid lowering drugs. This includes both metabolomics ("Turning to the metabolomics data revealed that 110 metabolites were differentially expressed between STARNET individuals currently taking cholesterol lowering medications versus those not on such medications (Supplementary Table 4)"); and coexpression networks.

We have investigated what drugs these individuals were taking, and found that the only class of drugs represented were statins. While this might appear surprising for CAD patients perhaps in the USA or Europe, it is important to note that the STARNET subjects were recruited in Estonia,

where clinical practice differs somewhat from the USA and other European countries. We have updated both the sentences quoted above to reflect this:

“We divided the STARNET cohort into two groups: one comprised of individuals reported to take statins (N = 334, from here on referred to as “the statin group”), and one of individuals not reported to take any lipid lowering drugs (N = 182, from here on referred to as “the no statin group”).”

“In the metabolomics data, 110 features were significantly different (FDR<0.05) between STARNET individuals in the statin versus the no statin groups (**Supplementary Table 8**).”

3. This reviewer is puzzled by the use of apoE-deficient mice. These mice are not relevant for diabetic dyslipidemia and for human pathophysiology. Indeed, ” in contrast to GLD in STARNET, in the mouse the liver module was positively correlated with a variety of lipid traits. This difference may relate to the super-physiologic levels of plasma cholesterol in Apoe-/- mice (1000 mg/dl versus 300 mg/dl as typically observed in humans with familial hypercholesterolemia), as the lack of Apoe in these mice results in near blockage of VLDL and LDL particle uptake primarily in the liver.”. Despite this, the Apoe-/- mouse intercross populations were used to validate and prioritize the KDGs. This weakens the study.

We thank the reviewer for bringing this important point to our closer attention. We have decided to remove this section as we agree with you and also Reviewer 3 that Apoe -/- is a poor model for dyslipidemia in humans with its excessive levels of plasma cholesterol. Instead we have analysed expression data from the Hybrid Mouse Diversity Panel⁴ bred on to wild-type chow-fed mice as well as to the atherosclerosis-prone Apoe-Leiden mice both with more moderate levels of plasma cholesterol similar to humans. In both of these datasets, we could confirm that the GLD mouse orthologue-genes are positively correlated with blood glucose and some GLD genes which are negatively correlated to cholesterol (see heatmap below). Thus, it appears that replacing the Apoe -/- mouse with these HMDP strains provide a much better match to humans. Still the correlations to plasma lipids are weaker than what we observed in STARNET. There are many plausible explanations related to differences in the molecular control of plasma cholesterol levels between mice and humans. A major difference is that blood cholesterol in mice is maintained mainly by HDL (LDL in humans) and unlike humans, mice carry nearly half of their blood cholesterol in apoB48-containing particles (in humans > 95 % are in apoB100-containing particles). As it is likely that at least some GLD module genes (e.g., *PCSK9*, *HMGCR*, *SREBF2*) control plasma cholesterol through regulating hepatic levels of the LDL receptor, such regulation may have less impact in mice since HDL is taken up by the liver mainly through scavenger receptors (i.e., Scavenger Receptor B1 (SR-B1)) and apoB48 does not bind the LDL receptor but to low-density lipoprotein receptor-related protein (LRP) through apoE. None of these genes (SR-B1, LRP) are present in the GLD module. The fact that mouse liver synthesizes apoB48 in the liver (human only synthesise apoB100) may also impact VLDL/LDL assembly and release to the circulation as it has been shown that the LDLr, by recirculating to the membrane in droplet vehicles (a process inhibited by PCSK9), targets newly synthesized apoB100-containing VLDL/LDL particles⁵ whereas apoB48-containing VLDL/LDL-particles

synthesized in the mouse liver will escape such control. In addition, mice lack *CETP*. Unfortunately, there is no perfectly matched mouse model of human cholesterol metabolism.

We have added a sentence on this as a possibility to explain some discrepancy between the human and mouse GLD module results in relation to plasma cholesterol in the Discussion section in the revised version of the manuscript (lines 438-449).

“While total and LDL cholesterol were not reduced, as we would predict in humans, this may have been expected here since the metabolism and regulation of plasma cholesterol in mice is quite different from that in humans. Most notably, wild-type mice carry a majority of their plasma cholesterol in HDL particles, not in LDL particles as in humans. As a consequence, mice have low and stable levels of LDL cholesterol unless they have been genetically engineered to model hyperlipidemias. In particular, it has been shown that statins do not alter LDL levels in normolipemic mice.⁶ Since our model predicts that the target of statins, *HMGCR*, is downstream from *LSS*, a similar unresponsiveness to inhibition of *LSS* could be expected. Nevertheless, it is clear that inhibition of *LSS* caused a significant perturbation in cholesterol synthesis. This is consistent with previously published results that inhibition of *LSS* measurably reduced the rate of LDL cholesterol synthesis in rats and mice, even though no change in the total levels of serum LDL cholesterol was reported⁷.”

4. How specific is the LSS inhibitor?

We extracted RNA from the livers of 12 mice from the experiment (6 in the LSS inhibitor, BIBB515 group and 6 in the control group). From this, we find at an FDR <5%, there are 1714 genes that are differentially expressed between the 2 groups. Of these 1714, 14 are overlapping with the gold module (which has 50 mouse orthologues expressed) which is more than is expected by chance (fisher's exact test p-value: 0.005922 and OR: 2.46). The 14 genes are *Scd2* *Tm7sf2* *Aldoc* ***Mvk*** ***Cyp51*** *Pold2* *Dhcr24* ***Idi1*** ***Tmem97*** *Fdft1* *Fads2* ***Dhcr7*** *Nsdhl* *Lss* (in bold are the 6 genes which are KDs in STARNET). We have added these new data to the Results section in the revised version of the manuscript (lines 369-372):

“Differential expression analysis showed a total of 1,714 genes whose expression was significantly altered in mice treated with BIBB-515 (FDR < 0.05). Of these, 14 were orthologs of GLD genes, a substantially larger overlap than would be expected by chance (odds ratio: 2.46, Fisher's exact p-value 0.0059).”

5. Overall, the phenotypic description of the STARNET cohort is weak.

As this is not the first publication of the STARNET cohort, we felt it was not necessary to include a detailed phenotype description. However, in the Results section we have now added some additional description of STARNET with clear reference to earlier studies (lines 97-105)

“Our principal data resource was STARNET, a cohort recruited at the Tartu University hospital in Estonia including patients diagnosed with CAD who were eligible for coronary artery bypass grafting (CABG) surgery. We analyzed the extensive omic datasets provided in the STARNET

cohort, including genotypes, gene expression levels from the arterial wall, blood and metabolic tissues, which is combined with a variety of clinical data that in addition to age, gender, body-mass index (BMI) and ethnicity also include standard blood biochemistry including plasma lipids and blood glucose levels, a full Nightingale metabolomic profile⁸⁻¹⁰, and also current and prior diagnoses and medications. A more detailed profile of this cohort has been published previously¹¹.”

6. The authors state that ”SNPs proximal to HMGCR, the enzyme targeted by statins, have been shown to decrease LDL and cause increased T2D risk.” To my understanding, the studies have found associations, but no real causality has been proven.

We had based this off of the Swerdol et al paper: “HMG-coenzyme A reductase inhibition, type 2 diabetes, and bodyweight: evidence from genetic analysis and randomised trials” (Lancet 2018) which showed that rs17238484-G was associated with increased T2D risk using a Mendelian Randomization approach. This study supports a causal relationship but only partially explains the risk of T2D. In the Result section of the revised version of the manuscript (lines 71-73), we have added to following sentence to be clearer.

“Moreover, MR studies of SNPs proximal to HMGCR, the enzyme targeted by statins, suggest that increased risk of T2D noted with statins could at least partially be explained by HMGCR inhibition¹². ”

Reviewer 3

The authors elucidated the perplexing juxtaposition between LDL and T2D in CAD patients. They used systems biology tools for the integration of several publicly available datasets and identified a small subnetwork that might be a key module for regulation of both cholesterol and glucose metabolism. They also performed additional analysis using Apoe^{-/-} mice model and tried to validate their finding in a mouse experiment. However, the results of the analysis in mouse model is not convincing. Overall, the manuscript is very well written, and the story line is quite clear. However, the authors should put additional attention in the validation of their results.

I have number of comments for further improvement of the manuscript:

Major concerns:

1. The authors should present the changes in the plasma or tissue more carefully. It is not clear in the manuscript. Figure 1A is also misleading. I suggest the authors clearly present the tissue/plasma transcriptomics and metabolomics data is used. It is so much confusing in the paper.

We thank you for this comment and your constructive feedback on our paper. Throughout the main text updates have been made to be clearer about when we used transcriptomics (from the 7 tissues including blood) and when we used metabolomics (only from blood plasma). For all tissues, we used whole tissue transcriptomics (i.e., RNA sequencing). In addition, we have updated the schematic of the study design in Figure 1 that we believe should help to improve clarity how different datasets have been used throughout the study.

2. This study employed a lot of computational analysis done by a series of packages and the key results are heavily dependent on these analyses. Therefore, it would be essential and appreciated if the author release the script for all the computational analysis so that the readers could generate the results presented in the paper.

Based on this comment, we have added Supplementary File 1, which contains instructions and sample code on how to run all analyses mentioned in the text.

3. The module identified by mouse dataset is different from the ones from human cohort, and the author claimed that the reason might be ‘the super-physiologic levels of plasma cholesterol in Apoe^{-/-} mice’. However, in their experimental validation, they fed the mice with chow diet but the correlation with plasma cholesterol level is still negative. This inconsistency needs to be properly discussed. The authors may even remove the analysis of Apoe^{-/-} mice model since it does not fit in the paper. I also suggest the use of liver transcriptomics data presented in Jake Lusis’ Cell Systems paper for providing additional support for their findings.

We have decided to remove the Apoe^{-/-} section as we agree with you and Reviewer 1 that this was a confusing section that didn’t add significantly to our study. Thank you for this comment.

As suggested, we have instead analysed hepatic RNA sequence data from the Hybrid Mouse Diversity Panel provided by the Lusis lab⁴. We used both the wild-type chow-fed mice and the atherosclerosis-prone Apoe-Leiden mice, both with more moderate levels of plasma cholesterol similar to humans. In both these datasets, we could confirm that the GLD mouse orthologue-genes are positively correlated with blood glucose and some GLD module genes which are negatively correlated to cholesterol (see heatmap below). Thus, it appears that replacing the Apoe -/- mouse with these HMDP strains provides a better match to humans. Still, the correlations to plasma lipids are weaker than what we observed in STARNET. There are many plausible explanations for this related to differences in the molecular control of plasma cholesterol levels between mice and humans. A major difference is that blood cholesterol in mice is maintained mainly by HDL (LDL in humans) and unlike humans, mice carry near half of their blood cholesterol in apoB48-containing particles (in humans > 95 % are in apoB100-containing particles). As it is likely that at least some GLD module genes (e.g., *PCSK9*, *HMGCR*, *SREBF2*) and *NPC1L1*) control plasma cholesterol through regulating hepatic levels of the LDL receptor, such regulation may have less impact in mice since HDL is taken up by the liver mainly through scavenger receptors (i.e., Scavenger Receptor B1 (SR-B1)) and apoB48 does not bind the LDL receptor but to low-density lipoprotein receptor-related protein (LRP) through apoE. None of these genes (SR-B1, LRP) are present in the GLD module. The fact that mouse liver synthesizes apoB48 in the liver (human only synthesise apoB100) may also impact VLDL/LDL assembly and release to the circulation as it has been shown that the LDLr, by recirculating to the membrane in droplet vehicles (a process inhibited by PCSK9), targets newly synthesized apoB100-containing VLDL/LDL particles⁵ whereas apoB48-containing VLDL/LDL-particles synthesized in mice will escape such control. In addition, mice lack *CETP*. Unfortunately, there is no perfectly matched mouse model of human cholesterol metabolism.

We have added a sentence on this as a possibility to explain some discrepancy between the human and mouse GLD module results in relation to plasma cholesterol in the Discussion section in the revised version of the manuscript (lines 387-390):

“The lack of response to lipid-lowering drugs in mice may have several explanations, including hepatic synthesis and assembly of apoB48-containing lipoproteins (apoB100 in humans)⁵ and having HDL as its main carrier of blood cholesterol (LDL in humans).”

4. The authors analysed gene expression data from 7 different tissues and reported LSS is the master regulator of the glucose and lipid metabolism based on liver tissue analysis. Similar analysis have been done in a manuscript by Lee et al, Mol.Syst.Bio and LSS have been reported as the regulator of lipid and glucose metabolism in almost all tissues analysed in the paper. The authors should be careful in reporting the role of LSS in all tissues since its role is not specific to liver.

We are uncertain what reference this is, but assume this is the Lee article in *Mol Syst Bio* the reviewer refers to¹³ on liver-specific targets using co-expression networks in various human tissues. On page 13 of the paper, we identified a link to the identified networks where

relationships of LSS with other genes in various tissues can be identified (<http://inetmodels.com/>).

Indeed, it is shown that LSS is expressed in a variety of tissues other than liver. The most commonly discussed role of LSS is in the eye. Mutations in LSS increase formation of cataracts¹⁴.

In STARNET the expression pattern of LSS across tissues, including some recently added primary macrophages and related foam cells derived in vitro, shows robust expression in liver and fat (see below panel). The modules that contain LSS in these tissues (VAF turquoise and SF turquoise) are not among the 20 modules we report as related to lipid and glucose metabolism, and are not enriched for any Gene Ontology (GO) terms related to cholesterol, insulin, or blood glucose.

This considered, the known importance of liver in lipid and glucose regulating processes and the gene content (PCSK9, HMGCR, SREBPs etc) of GLD module, the most reasonable assumption is that the GLD module is a key regulator of lipid and glucose metabolism in the liver, a notion further supported by the experimental validation in mice.

However to acknowledge that LSS also has been shown to have roles in other tissues, we have updated the main text including the Lee article, which has been added in the revised version of the manuscript in line 301-303:

“This gene [*LSS*] has previously been identified as an important driver of various disease-related processes in liver as well as in other tissues¹³, but it has not previously been associated with glucose metabolism.”

5. The last part of the results section should be rewritten carefully. The model predictions and the experimental data should be presented clearly in Results section and discussed in the last part of the paper.

The key finding in this study is the GLD module and its positive correlation with cholesterol metabolism and negative correlation with glucose metabolism. But the experimental validation in mice failed to demonstrate that.

In addition, the authors claimed in line 335 that ‘our model predicted that inhibiting LSS would down regulate pathways associated with lipid metabolism that would result in increased cholesterol and increased glucose levels’, which is very confusing since the correlation to cholesterol was suggested to be positive.

The authors simply stating that the validation of the study is not performed properly. Please can you clarify this statement. Such statement is quite disappointing.

“We found that LSS in our mouse model significantly perturb the GLD network as our model predicts, and that while glucose levels were increased as predicted, total cholesterol was not reduced as expected. That cholesterol levels were not reduced may largely reflect the inadequacy of mouse as a model of cholesterol control in humans”.

Thank you for pointing out the sub-optimal way we had discussed the mouse validation experiment. While the key finding of our study is indeed that in humans the GLD module regulates lipid and glucose levels in opposite directions, data from the “perfect” mouse model mimicking human plasma cholesterol metabolism are simply not available (see also response to comment 3). Thus, demonstrating the effects of the GLD module on plasma cholesterol wasn’t the purpose of the mouse validation experiment. Instead, we argued that since inhibition of LSS repeatedly has been shown to reduce plasma LDL cholesterol levels, the mouse model validation was primarily intended to examine to what extent LSS: 1) perturbs the activity of the GLD module and 2) if such perturbation was accompanied with an increase in glucose levels. Indeed in the mouse validation experiment, both these hypotheses were clearly demonstrated to be true.

To clarify the LSS validation experiment data, we have rewritten these paragraphs both in the Results (lines 354-399) and the Discussion (lines 434-449) sections. These sections are also copied below:

“To test the effects of lanosterol synthase (*LSS*) as a GLD module key driver involved in regulating both cholesterol levels and glucose metabolism, we inhibited the enzyme in B6 mice and observed clinical and transcriptional effects. Our model predicted that inhibiting *Lss* would reduce the activity of the GLD module, resulting in decreased production of LDL cholesterol and increased glucose levels. Pharmacological inhibition of *Lss* was achieved by using BIBB-515, a compound that selectively inhibits *Lss* enzymatic activity. BIBB-515 was developed as a non-statin cholesterol reducing medication and has been shown to significantly disrupt hepatic cholesterol synthesis in rats and mice and rapidly decrease LDL cholesterol levels in hamsters. These earlier findings reinforce our model’s prediction that inhibiting *Lss* lowers LDL cholesterol levels. In contrast, effects of *Lss* inhibition on hepatic gene expression and associated

blood glucose levels have never been investigated. Our model predicted that inhibiting Lss should significantly alter expression of downstream GLD module genes, accompanied with increased levels of blood glucose. To validate these model predictions, we treated C57BL/6J (B6) mice with a diet containing BIBB-515 (55 mg/kg) for 10 days, and compared the plasma glucose and hepatic gene expression profiles to mice fed a control diet (n = 11 per group). Differential expression analysis showed a total of 1,714 genes whose expression was significantly altered in mice treated with BIBB-515 (FDR < 0.05). Of these, 14 were orthologs of GLD genes, a substantially larger overlap than would be expected by chance (odds ratio: 2.46, Fisher's exact p-value 0.0059). Targeted measurement of fold-change in gene expression by quantitative PCR (qPCR) showed increased expression of GLD module genes involved in cholesterol synthesis in mice treated with BIBB-515: *Lss* (FC: +1.9, *P*: 0.004), *Dhcr7* (FC: +1.9, *P*: 0.0009), *Idi1* (FC: +1.7, *P*: 0.04) and *Hmgcr* (FC: +2.4, *P*: 0.0005) (**Fig. 5A**). Other GLD module genes that also showed increased expression in response to Lss inhibition were *Pcsk9*, *Mvd* and *Rdh11* (**Supplementary Fig. 6**). *Adipor2* was the only GLD module gene analyzed by qPCR that did not show a significant change in expression (*P*: 0.06). In agreement with our STARNET models demonstrating correlation between expression of GLD module genes blood glucose levels, blood glucose level increased 14.5% in response to Lss inhibition by BIBB-515 (*P*: 0.0006) (**Fig. 5B**). Correspondingly, the expression of key gluconeogenesis genes, phosphoenolpyruvate (*Pck1*) and glucose-6-phosphatase (catalytic subunit *G6pc*), increased two- to three-fold in response to the Lss inhibition (FC: +3.0 and 2.1; *P*: 2E-7 and 0.0006, respectively) (**Fig. 5C**). In contrast, we observed no significant effects on levels of non-HDL plasma cholesterol in response to the Lss inhibition (*p*=0.17). This was not entirely unexpected given previously reports showing minimal response to lipid-lowering drugs in wild-type normolipemic mice.⁶ The lack of response to lipid-lowering drugs in mice may have several explanations, including hepatic synthesis and assembly of apoB48-containing lipoproteins (apoB100 in humans)⁵ and having HDL as its main carrier of blood cholesterol (LDL in humans). On this note, the Lss inhibition did increase the levels of HDL cholesterol (FC: +10.0%, *P*: 0.003) and in parallel total plasma cholesterol (FC: +9.4%, *P*: 0.014) (**Supplementary Fig. 7**). Since the levels of HDL cholesterol are normally negatively correlated with LDL cholesterol, the increase in plasma HDL represents a perturbation of cholesterol metabolism in the predicted direction. Overall, the results from the experimental validation are consistent with our model's predictions that inhibiting Lss should perturb the GLD module genes and increase blood glucose levels. In addition, considering previous reports establishing that inhibition of Lss disrupts hepatic production of LDL cholesterol particularly in other species than mice and the reciprocal increase in HDL levels observed in our experiments, the model prediction of decreased LDL was at least partly fulfilled.”

“In our mouse study, given liver *LSS* is a KDG of GLD, our model indicated that inhibition of liver *LSS* would alter the GLD network state, which in turn would lower cholesterol levels and increase glucose levels. We found that inhibition of hepatic Lss in our mouse model significantly perturbed the GLD network and increased glucose levels, as our model predicted. While total and LDL cholesterol were not reduced, as we would predict in humans, this may have been expected here since the metabolism and regulation of plasma cholesterol in mice is quite different from that in humans. Most notably, wild-type mice carry a majority of their plasma cholesterol in HDL particles, not in LDL particles as in humans. As a consequence, mice have low and stable levels of LDL cholesterol unless they have been genetically engineered to

model hyperlipidemias. In particular, it has been shown that statins do not alter LDL levels in normolipemic mice.⁶ Since our model predicts that the target of statins, *HMGCR*, is downstream from *LSS*, a similar unresponsiveness to inhibition of *LSS* could be expected. Nevertheless, it is clear that inhibition of *LSS* caused a significant perturbation in cholesterol synthesis. This is consistent with previously published results that inhibition of *LSS* measurably reduced the rate of LDL cholesterol synthesis in rats and mice, even though no change in the total levels of serum LDL cholesterol was reported⁷.”

6. There is also a fundamental issue in the analysis performed in the paper. I do not think an intervention increasing the plasma glucose can provide efficient treatment strategy. The authors should discuss this in the paper. In my opinion, inhibition of LSS in all tissues should not be presented as a good strategy. My question to the authors is if they would run a clinical study using LSS inhibition?

We are not suggesting *LSS* inhibition as an intervention, but rather are using it as an attainable way to perturb the GLD module in order to validate experimentally the results of our systems analyses, which include the identification of *LSS* as a novel key regulator of these pathways. We agree with your comment in that having an intervention that increases glucose and lowers cholesterol would not be ideal (even though that is also a side effect of statins). If we were trying to develop an effective clinical intervention, we would also evaluate other key drivers of the GLD module as potential targets, such as *DHRC7*. However, we believe this undertaking is outside the scope of the current study.

Minor concerns:

1. It might be very obvious to the authors, but ‘OR’ need to be annotated as ‘Odds Ratio’ when first time used.

Thank you for pointing this out to us. This has been revised in the new version of the manuscript.

2. In line 167, the author mentioned the GTEEx cohort is separated into two groups based on age, which is very strange to me and need justification. The relationship between age and the biological question in this study is not sufficiently discussed.

We had divided the GTEEx cohort into 2 ages as ~40% of this cohort builds on individuals who suffered heart disease, but as individual phenotype data are missing, it is unclear who. Thus, we argue that by dividing the cohort into two groups based on age, we could assume with some confidence that the younger group are heart-healthy.

According to this reviewer’s wishes, we have further outlined the reason for dividing the GTEEx cohort based on age in the revised version of the manuscript on lines 191-193:

“In order to partially control for the lack of individual phenotype data in GTEEx, we divided the GTEEx cohort into two groups based on age, so that the younger group would be more likely to be heart-healthy (see Methods).”

3. In line 208, I would suggest to use ‘significantly changed’ instead of ‘differentially expressed’ since metabolites are not expressed by the cells.

Thank you for this comment, this has been revised in the new version of the manuscript.

Reviewer #4:

This is an interesting manuscript, which leverages integrative approaches in STARNET (and other) cohort to identify a liver gene co-expression network, and therein the LSS gene, underlying both hepatic lipid and glucose metabolism pathways.

I believe the manuscript can be improved by (1) explaining the integrative analyses, (2) making all results/data available to the reader, and (3) performing additional experiments to corroborate the Authors' key conclusions. In its current form the manuscript is difficult to follow, even for a reader (as myself) who is familiar with the kind of integrative approaches and network-based analyses presented here. In my detailed comments (below), I highlight several areas for improvement, and I suggest ways to better the presentation of the results. I also believe that some methodological details and statements in the manuscript should be better justified, especially for a non specialised audience. The in vivo validation experiments can be also strengthened.

Thank you for your comments. We have worked hard on editing the text and adding in more experiments to strengthen the paper.

Detailed comments

The explanation of the data-driven computational strategy and integrative network approaches presented in Fig. 1 and in the text (page 5, first paragraph) can be improved by reporting (within the figure) the number of DE genes, networks (replicated and non-replicated), and KDG identified at each step of the analysis, and which crucial statistical thresholds were used at each step. This will help the reader to assess how the KDE are selected for in vivo validation starting from a transcriptome-wide analysis, highlight the support provided by integration of independent datasets, and outline the crucial steps in the KDE prioritization strategy.

Thank you for this comment, we have substantially altered Figure 1 and the text (lines 106-123) in response. The original version of the figure was misleading, as it gave the impression that these are steps in one single analysis that starts with transcriptome data and ends with key driver genes, as described in this comment. It is more accurate to present them as distinct analyses, each of which provides a different lens on the biology of lipid and glucose regulation.

“To uncover the molecular drivers of plasma LDL and blood glucose levels, we employed an integrative network approach utilizing multiple different data captures as depicted in **Fig. 1**. The overall goals were to systematically characterize all gene-clinical trait associations, all clusters of coexpressed genes, and the associations of those gene clusters with clinical trait data and metabolomics. To achieve this, we performed a series of three analyses. First, we performed differential expression analysis using STARNET gene expression data across 7 tissues and clinical information, identifying a total of 12,872 unique genes across 7 tissues whose expression was correlated with at least one glucose or lipid related clinical trait. Second, we performed coexpression clustering analysis on all 7 tissues, identifying 140 tissue-specific gene clusters. Comparing these clusters to the results of the differential expression analysis, we found a total of

20 clusters of coexpressed genes across 3 tissues enriched for these glucose and lipid correlated genes. We identified one of these modules in particular as a liver glucose and lipid determining (GLD) module and verified its presence in two additional human transcriptomics datasets and one mouse transcriptomics dataset. Finally, we constructed four different probabilistic causal network models integrating transcriptomics, genomics, clinical traits, and metabolomics. We then mined all four of these networks to identify 30 candidate master regulator genes for the GLD module, and validated the physiological relevance of the top candidate with respect to plasma lipid and glucose control using an *in vivo* experimental mouse model system.”

The use of terminology DE signature and DE genes should be used cautiously to avoid confounding the reader. Typically, DE refers to a contrast between two states (e.g., case/controls) and the general reader will assume canonical differential expression analysis (or fold change analysis) between two conditions. Here “DE signatures” refer to genes whose expression correlates with a given trait. Therefore, sentences like “the largest DE signature, comprised of a total of 4,653 unique DE genes” can be confusing. To improve readability throughout the whole manuscript (and supplementary materials), I suggest to use “gene signature correlated” with trait or “DE signature correlated” with trait, and to specify “correlated with” all times this is required to clarify the nature of the DE gene signatures.

For each DE signature, the complete list of genes correlated with any trait in any tissue should be made available to the reader. In addition, a detailed summary of the overlap between the different DE signatures across tissues should be also made available. (Ideally, a small web portal to browse all DE signature genes and their correlations with traits will really help the reader to access and browse these extensive results.)

Thank you for this point, we have clarified the text to refer to the DE signature more explicitly as trait correlated DE gene signatures. We have also added Supplementary Table 1, which lists all genes significantly associated with glucose and lipid traits in the 7 tissues.

Related to the previous comment, in Figure 1 I assume the numbers in between the colors (e.g., in LIV, number 5 between 1243 and 1116) represent few DE signature genes that are too small to be evident in the figure. Even at high magnification, I cannot see colors (trait) associated with these numbers. Are the 5 DE genes in LIV (between 1243 and 1116) correlated with bl.glucose? The same question applies to the 2 DE genes in VAF.

Thank you for this comment, we have updated the figure to make the colors associated with these numbers clearer.

“We ran gene ontology (GO) enrichment analysis for each DE signature in each tissue to assess their biological relevance”. Which method (tool) was used for this analysis? Most critically, what is the gene set background used in this enrichment analysis? Is the gene set background tissue specific? Does the gene set background account for the number of genes expressed in each tissue or across all tissues? These details are missing in the methods and are important to assess whether the GO enrichments are specific to given tissues.

We have updated our methods section to include the details of the GO enrichment for the DE signature set. We used the background set as all genes that were expressed in that tissue and have clarified the methods to include this (lines 576 – 583):

“Enrichment Analysis

We annotated each gene with its associated GO categories and KEGG pathways using the goseq package in R version 3.1.0. We performed the enrichment analysis using the topGO package in R version 3.1.0, which performs a Fisher’s Exact Test on each annotation category using the provided foreground and background gene lists and uses the Benjamini-Hochberg procedure to select significantly enriched annotations. We used as our foreground the list of significant trait correlated DE genes in each tissue, and as our background the list of all genes expressed in the same tissue.”

It will help if the same trait nomenclature is used consistently between the main figure and Supplementary Table 1 (and elsewhere throughout the whole manuscript as well as in supplementary materials).

Thank you for this comment. We have updated Figure 2, Figure 4, Supplementary Figure 2, Supplementary Table 1 (now ST2), and all mentions of the clinical trait data in the text to use the same set of trait names, as well as to make these trait names more readable.

In the sentence “Genes in the glucose and lipid trait DE signatures were significantly enriched for 586 common GO terms, reflecting a significant enrichment of common GO terms (Fisher’s exact test OR: 108.8 $P < 3E-324$), reflecting a high-degree of interaction between pathways associated with lipid and glucose biology”, are the Authors referring to DE signatures across all tissues? This should be specified here.

Thank you for this comment, we have updated the text to be more clear of what was done. Please see lines 132-138 for a revised version of these sentences, copied below. As we had run each DE in each tissue independently and corrected each tissue for multiple testing ($FDR \leq 0.05$), we used the full set of genes expressed in each specific tissue as the background genes for each tissue-specific analysis.

“The glucose and lipid trait correlated DE signatures had highly overlapping GO terms across all traits and all tissues, with 586 GO terms shared in common across all. This represents a highly significant sharing of GO terms compared to what would be expected for independent lists (Fisher’s exact test Odds Ratio (OR): 108.8 $P < 3E-324$), reflecting a high degree of interaction between pathways associated with lipid and glucose biology across tissues (**Supplementary Fig. 1A, Supplementary Table 2**).”

Beyond those related to glucose and lipid traits, are there other significant GO terms overlaps? A systematic analysis of all GO overlaps should be reported (supplementary).

GO enrichments for all traits in all tissues are reported in Supplementary Table 2 (formerly ST1). The only traits we considered for this study are glucose and lipid traits. We consider detailed profiling of other traits to be outside the scope of this work.

In addition, given that these 586 common GO terms are detected for the glucose and lipid trait DE signatures across all tissues, an appropriate background gene set used in the GO enrichment analysis should be the common set of genes expressed in all tissues. To reinforce this initial observation (Suppl. Fig. 1) the authors should test how the GO enrichment results are affected by the background gene set used in the analysis (tissue-specific vs across-tissue).

We believe it is not appropriate to restrict the background for the tissue-specific analyses further than the list of genes expressed in that single tissue, since a GO term that is enriched in multiple tissues does not necessarily represent an identical list of genes in all tissues. The only analysis that looks across multiple tissues is the test of overlap in GO terms across multiple tissues, for which we used the list of all GO terms enriched in any tissue as the background.

“We identified 20 modules across the liver (N= 6), VAF (N= 5) and SF (N=9) that were enriched for both lipid and glucose DE signatures...”. The full list of WGCNA modules, the genes therein (including those that are part of the DE signatures), the complete GO enrichments for all modules and the genes contributing to the GO enrichment should be provided to the reader (Supplementary or via web portal).

We have updated Supplementary Table 3 to include the full set of modules, and Supplementary Table 1 contains the full list of DE signatures for each gene in each tissue.

How was the direction of the correlation between the modules and the trait been assessed? In Suppl. Table 2 and if possible in Fig. 2B too, it will help to have clear representation of the degree to which these modules are correlated (positively or negatively) with each trait. The correlation is reported in Fig. 2B (red-blue color scale), yet it appears that in most cases the R2 are rather small. E.g., for the LIV gold, it seems that none of the correlations are greater than 0.3, which is R2 less than 10%. Given that the 1st PC used to assess these correlations explains ~52% of variation and R2 is less than 10%, are these correlations biologically meaningful? Also, what about the second (and third) PC? These PCs capture less variance of the module gene expression variation, but might show much higher correlations with the traits. Lastly, it will be also useful to have for each gene in the module, the raw gene-traits correlation statistics, in particular for the GLD module.

We first calculated the 1st principal component of each module as a proxy for the group of genes in that module and correlated PC1 to all phenotype traits and corrected for multiple testing using BH method (FDR < 0.05). While we understand the concern expressed here and elsewhere about correlations in the range of 0.1-0.3 seeming weak, all correlations we report are highly significant after multiple test correction. Additionally, they are similar in magnitude to expression-phenotype correlations observed in this dataset for many genes with well-established biological significance in cholesterol metabolism, such as liver HMGCR (r with LDL

cholesterol: -0.17; raw p-value = 0.0001) or liver PCSK9 (r with LDL cholesterol: -0.09; raw p-value = 0.05).

Thank you for the suggestion to look at PC2 & PC3. Some of the modules shown in Supplementary Table 2 (now ST4) do have significant associations to lipid and glucose traits in PC2 and PC3, some of which are not significant in PC1. However, overall the higher PCs decrease in biological importance: the number of significant correlations, the magnitude of correlations, and the significance of correlations all decrease with higher PCs. We have added Supplementary Table 1 containing gene-trait correlations in all 7 tissues, and Supplementary Table 5 containing all trait correlations with principal components 1-3 for all glucose and lipid associated modules.

The correlations between the metabolites and PC1 of GLD module are weak. From Suppl. Table 4, the maximum R2 with any metabolite is ~10%, and given that PC1 explain 57% of the variance of GLD module, these association are very modest, and hence their biological significance is questionable. For instance, among the results highlighted in the text, the “positive correlation with Glucose” has $r=0.17$ (i.e., $R^2 = 3\%$) and it is not differentially expressed between STARNET individuals taking cholesterol lowering medications vs those not on such medications. Can these associations with metabolites be independently verified?

The metabolite correlations are harder to independently verify as we don't have access to other datasets with gene expression and detailed metabolomics. We do note that while the correlations are not very strong, they are highly statistically significant and that the metabolomics correlations replicate the findings in STARNET liver and the morbidly obese cohort both in the broad categories of correlation but also their directionality.

“We constructed coexpression networks for each tissue, identifying 40, 56 and 32 modules in liver, subcutaneous fat and visceral fat, respectively.” Again, all data should be made available in full to the reader, including detailed statistics on the overlay with the GLD module. Related to this, are there other modules (in addition to GLD) that overlapped with the modules identified in 600 obese patients?

We have updated the methods to include the detailed statistics on the overlay with the GLD module to be clearer and as per Reviewer 3's suggestions as well we have added Supplementary File 1 containing sample code.

“...correlating the first principal component of this liver module with clinical metabolic traits measured in these 600 individuals found positive correlations with glucose levels and a negative correlation with plasma LDL levels ($FDR < 5\%$)”. More details are also needed here, and in particular with respect to the magnitudes of these correlations. All data should be made available to the reader. Do the authors detect the same metabolite-module correlations as in Supplementary Table 4? A detailed metabolite-specific comparison should be carried out here.

Thank you for this comment, we have added Supplementary Table 10 containing this data.

“the 12 tissues currently represented in GTEx with sufficient sample sizes to construct coexpression networks”. Why sample sizes greater than or equal to 30 is sufficient to construct coexpression networks? Is this sample size based on power calculation or other empirical evidences?

Coexpression is calculated using Pearson’s correlation, which requires a minimum sample size of approximately 30 to be reliable. We have added the following sentence to the methods (lines 593-595):

“We chose to perform coexpression analysis only on tissues with at least 30 samples, as this is the approximate minimum number of samples required to compute a reliable correlation coefficient between expression levels.¹⁵”

Since the Authors used “the STARNET cohort, given that CAD provides a more extreme context in which metabolic traits such as cholesterol levels are more significantly perturbed”, and replicated their findings in another disease cohort (600 morbidly obese patients undergoing bariatric bypass surgery), what is the rationale to seek further replication in GTEx? Indeed, replication in tissues from healthy subjects (GTEx) plays against the initial Authors’ rationale (with which I agree) to consider cohorts where cholesterol levels are more significantly perturbed.

By looking at the GTEx data we were able to assess if there were more tissues in which these genes clustered together. Also, as per the GTEx information they state that while these individuals died not due to health-related issues, 40% of the patients have CAD, however, they do not state which. Dividing the GTEx data into two groups of older and younger as a proxy for CAD we still find these genes are correlated in both groups.

“Across the 58 liver and 38 adipose modules identified, only a single module was identified as enriched for GLD module genes at an FDR <5%”. All data should be made available to the reader. What is the overlap with the modules identified 600 morbidly obese patients and overlapping with GLD? A detailed three-way comparison would be useful here. Are there any other modules (apart from GLD) that have significant overlap between STARNET, 600 morbidly obese patients and the ApoE-/- mice on a high fat diet?

As per the recommendations from Reviewer 1 & 3, we removed the data from apoE-/- mice as we agree that they don’t fully represent the best biological model of particularly cholesterol metabolism in humans. We have instead looked at the HMDP mice fed a normal, chow diet and those on an atherosclerotic apoE leiden background. Unfortunately for these data the gene expression does not follow a scale-free assumption (the major assumption for constructing coexpression modules) and so we did not explore the coexpression clusters of genes. However, we do note that the 60 GLD genes and their orthologues in these data are consistently correlated (FDR <0.05) with the traits of interest in the same direction.

In regard to other modules overlapping between the morbidly obese cohort & STARNET, in response to the reviewer's suggestion, we looked into this and found that many of the STARNET modules have significant corresponding modules in the obese cohort. This analysis is included as Supplementary Table 9. The gold module (GLD) has the highest OR overlap of all the modules.

Is “Consistent with STARNET, at an FDR <5%, the first principal component of this mouse liver module was positively correlated with glucose and insulin traits.” Again - more details and analyses are needed here. See also previous comments.

As noted above, we have removed the entire analysis of apoE^{-/-} mice, so this confusing sentence and the analysis it refers to are no longer present.

“The GLD module correlations to the lipid and glucose traits were also largely conserved across the different groups” and data presented in Fig. 3B; all specific correlations should be reported in detail. Again it appears these correlations are weak.

We have added these correlations as Supplementary Table 7.

“Leveraging DNA information allows us to determine causality, as we know that DNA must be upstream of gene expression”. I found this phraseology a bit unspecific. What is the meaning of “DNA must be upstream of gene expression”?

We thank the reviewer for pointing out the unclear sentence. We have clarified it to read as follows (lines 250-252):

“Leveraging DNA information allows us to determine causality, as we know that DNA must be causally upstream of gene expression – that is, a genetic variant can cause altered gene expression but a change in gene expression cannot modify germ line DNA.”

Expression QTL analysis STARNET liver samples: I suppose this analysis is limited to cis-eQTLs; what is the cis-window used to call eQTLs? Does the choice of the cis-window affect the number of detected eQTLs? Which FDR correction was used here? Does this FDR correction account for LD structure between the SNPs in the cis-window? Why did the Authors completely disregard trans-eQTLs here? The important role of trans-eQTL as drivers of known disease associations (including cholesterol metabolism) have been previously demonstrated (e.g., PMID:24013639).

In addition, as the focus here is on GLD, a co-expression module which should indicate co-regulation, this can be investigated by mapping trans-acting regulators of the GLD module, as previously shown (e.g., PMID:21572415).

We used the MatrixEQTL package which uses a linear model and a window of 1MB for calling cis eQTLs. As we had already published on both cis and trans eQTLs from STARNET, we were interested in using eQTLs for breaking the causal/statistical problem of A->B being the same as

B -> A. As such, we were only interested in using cis eQTLs, and also only interested in finding which genes had a cis eQTL. We agree with the Reviewer, that trans eQTLs are important but due to the computational complexity we were only able to use the cis values. The method we used took into account the SNPs being in LD in the 1MB region around the gene and reported only the top eQTL. We used an FDR threshold of 0.05 and this was calculated using permutations within that window.

“To model the relationship between these genes and the common regulatory control driving the coherence of this network,...”. What do the Authors mean by “the coherence of this network”?

To improve clarity we edited this sentence to read, “To model the relationship between these genes and the common regulatory control that defines this network as a coherent unit....”

Bayesian network analysis - a posterior probability of 0.3 was taken as a cutoff for returning results from the MCMC reconstructions. This posterior probability threshold seems very low, and here the reader has to believe this threshold is appropriate without any detail supporting this specific choice. Can the Authors better justify this choice (also, in context of successive removal of the “weakest link” (edge with the lowest posterior probability)?

This threshold is derived from a simulation study using this same methodology (Zhu et al. 2007, reference 42 in this manuscript), which tested a range of thresholds and determined that 0.3 gave the best balance between precision and recall. We have added a sentence to the Methods to clarify this (lines 630-632):

“These parameters are based on a previously reported simulation study, where using a posterior probability cutoff of 0.3 in 1,000 reconstructions was found to provide the best balance between precision and recall.^{16,17,”}

“This gene list was then further expanded using the PEXA algorithm, to incorporate known interactions from KEGG and PPI networks to 3163 genes (Fig. 4B)”. First, the pathway expansion analysis (PEXA) in ref.48 was implemented to leverage KEGG pathway database and PPI data as orthogonally generated sources of information to identify genes involved in processes associated with insulin signaling based on siRNA screening data sets. How was the PEXA algorithm used here? (This is not described anywhere in the manuscript). Second, Figure 4B is not very clear and its legend needs to include more details. The larger network contains >3000 genes, and in the figure only the GLD and Bisque modules are extracted to link genes involved in cholesterol and glucose pathways? I believe that this kind of presentation of results can be misleading since at this point in the manuscript one can identify many more pathways linked through 3163 genes (not just the cholesterol and glucose pathways). How are these two pathways extracted from the large network of 3163 genes?

We have updated the methods to include a more detailed section on the PEXA algorithm that we used and how we implemented it (lines 665-674). We highlighted the GLD module as that was

the module of focus in this paper and the bisque module was identified in the large network of 3163 as the module most connected to the GLD module and the only module downstream of it.

“PEXA Analysis

We applied the PEXA algorithm¹⁸ to select genes to include in our expanded networks. PEXA takes a seed list of genes, which in the original PEXA publication are genes identified by an siRNA screen, and uses KEGG pathway data and protein-protein interaction data to select related genes. For the GLD expanded network, our seed list contained all genes in the GLD module, and all genes whose expression in LIV was correlated with the first PC of the GLD module at an FDR < 5%. For the global liver network, we ran PEXA separately for each of the 28 modules in LIV identified by WGCNA, including the GLD module. For each run, we used as our seed list all the genes in that single module. We then combined the resulting 28 expanded gene lists to produce a single global expanded gene list.”

The data reported in Supplementary Fig. 5B are not very useful - what is the reader supposed to learn from the “hairball” network reported in panel B? This expansion of the GLD network to the global liver network does not seem to add much information here.

We expanded to the larger network to ensure more of an ‘unbiased’ approach to identify KDG for the GLD network. We agree that a ‘hairball’ is confusing and have included the Cytoscape files used to construct the image as Supplementary File 2, so that individuals can reconstruct the network and search for their own edges.

The list of identified KDGs is presented in Supplementary Table 7. In order for the reader to interpret and understand the data presented, it is important the Table’s contents are clearly explained. Since not all readers might be familiar with the KDA analysis and its parameterization, each table heading (downstream_signature_in_subnetwork, subnetwork_size, signature_in_network, network_size, signature_optimal_layer, etc.) must be adequately explained. In addition, a p-value = 0 (as for pvalue_whole, pvalue_subnet or pvalue_corrected_subnet) does not make sense, and this should be better replaced with the lower bound estimate of the p-value (e.g., it should be reported as “p<....”).

Thank you for pointing this out, we have changed the column labels of this table (now ST13) and removed a number of irrelevant columns. We have also removed the p-value column where it is invalid (the p-value measures the enrichment of GLD genes in the KDG’s downstream nodes, and is not valid for networks built using only GLD genes).

To validate and prioritize the KDGs identified across all networks constructed in STARNET, KDGs were predicted in the Apoe^{-/-} mouse data. What about the cohort of 600 morbidly obese patients (which was also used to replicate the GLD module)? It will be more powerful to seek replication of the KDGs in human cohorts (i.e., 600 morbidly obese patients) rather than in Apoe^{-/-} mouse alone. This might also help in better assessing the relative importance of DHCR7 vs LLS, as both these genes rank high and are predicted to be KDG in the Apoe^{-/-} mouse data (Table 1).

Thank you for this recommendation. Based on this we constructed the BN's in the morbidly obese cohort and ran KDA there as well.

In the obese cohort, for the network with the GLD only genes, the KDGs were SREBF2, FADS1, DHCR7, and CYB5B. And, for the expanded GLD network, the only KDGs for GLD was MMAB (2 separate probes). Of which DHCR7 and MMAB were found in the STARNET cohort. It is important to note that the morbidly obese patients are a different cohort that is phenotypically distinct from the STARNET cohort, and therefore might not be expected to behave the same. Also, the morbidly obese cohort was sequenced with custom micro array technology while STARNET was sequencing using RNAseq protocols which can carry their own error profiles. These issues are in addition to a generally low expectation of reproducibility of specific features of BNs at this sample size.¹⁹

“...the module that was most highly correlated with the GLD module was the bisque module (r value: 0.40,...”. How was this correlation calculated? Also, what is the null hypothesis in this case? (is $H_0 r \neq 0$)? Since all these modules are within the expanded GLD network, what is the average (expected) correlation to be used to assess the significance of each module correlations?

The correlations were calculated comparing the first PC of each module to each other. We ran a pairwise correlation and returned the paired student t-test. The null hypothesis is $r = 0$, as is standard for this test. We did this for all modules in the liver and the corrected the p-values using the BH method to adjust for multiple testing, returning only the results where the $FDR \leq 0.05$. We have added text clarifying this.

Something is wrong with the legend of Supplementary Table 8. “Liver modules correlated to GLD module at $FDR < 5\%$ and the number of edges to & from GLD genes in the expanded and global BNs. Type or paste caption here. Create a page break and paste in the Figure above the caption.” As mentioned earlier, and $FDR = 0$ does not make sense.

Thank you for catching this, we have fixed the legend for ST8 (now ST14).

“Importantly, key driver detection in the global BN revealed that LSS is predicted to be a KD of the bisque module as well as the GLD module”. Maybe I am missing something here, but where are the results (statistics) of the KDG analysis for LSS in the bisque module reported? In Supplementary Table 7 (GlobalBN tab) only KDG results referring to the gold module are reported and in Fig. 4B there are no clear data indicating LSS is a KD in bisque module.

Thank you for catching this, we have added to ST7 (now ST13) the results from the bisque KDA.

Given the suggested role of LSS as a master regular of networks underlying both lipid and metabolisms, in order to assess the contribution of LSS to liver gene expression and demonstrate its master regulatory role of GLD module, the Authors should carry out a comprehensive transcriptomic analysis (e.g., by RNA-seq) in the liver (rather than looking

at just 4 additional genes, with 1(out of 4) gene showing no significant changes in expression). Besides, this will also allow to assess specificity and magnitude of the predicted effect upon in vivo drug perturbation of LSS, including the suggested role of LSS as a KD for the bisque module. I believe qPCR data on 4 GLD module genes are insufficient to state that “altering Lss perturbed the GLD module genes”.

We extracted RNA from the livers of 12 mice for the experiment (6 in the LSS inhibitor BIBB515 group, and 6 in the control group). From this, we find at an FDR <5%, there are 1714 genes that are differentially expressed between the 2 groups. Of these 1714, 14 are overlapping with the gold module (which has 50 mouse orthologues expressed) which is more than would be expected by chance (fisher's exact test p-value: 0.005922 and OR: 2.46). The 14 genes are Scd2 Tm7sf2 Aldoc **Mvk Cyp51 Pold2 Dhcr24 Idi1 Tmem97 Fdft1 Fads2 Dhcr7 Nsdhl Lss** (in bold are the 6 genes which are KDs in STARNET). We have added a sentence to this paragraph (lines 369-372) describing this result:

“Differential expression analysis showed a total of 1,714 genes whose expression was significantly altered in mice treated with BIBB-515 (FDR < 0.05). Of these, 14 were orthologs of GLD genes, a substantially larger overlap than would be expected by chance (odds ratio: 2.46, Fisher’s exact p-value 0.0059).”

In addition, to further strengthen their conclusions, a dose dependent response analysis for both glucose and transcription in the liver would be beneficial here, although I understand this is an substantial experimental undertaking.

We agree that these experiments would be beneficial but that is a substantial undertaking. We feel that given this GLD module is so heavily replicated in independent data-sets and conserved across species that with the improved RNAseq analysis we have shown sufficient support to link this module. We believe this would be excellent follow up work (and a separate paper) to come.

I hope my detailed comments and suggestions will help the Authors improving the manuscript, and importantly to make it more accessible to the wider readership of Nature Communications.

**Reviewer:
Enrico Petretto, PhD
Duke-NUS Medical School (Singapore)
Imperial College London (UK)**

1. Willer, C.J. *et al.* Discovery and refinement of loci associated with lipid levels. *Nat Genet* **45**, 1274-1283 (2013).
2. Wheeler, E. *et al.* Impact of common genetic determinants of Hemoglobin A1c on type 2 diabetes risk and diagnosis in ancestrally diverse populations: A transethnic genome-wide meta-analysis. *PLoS Med* **14**, e1002383 (2017).

3. Manning, A.K. *et al.* A genome-wide approach accounting for body mass index identifies genetic variants influencing fasting glycemic traits and insulin resistance. *Nat Genet* **44**, 659-69 (2012).
4. Lusi, A.J. *et al.* The Hybrid Mouse Diversity Panel: a resource for systems genetics analyses of metabolic and cardiovascular traits. *J Lipid Res* **57**, 925-42 (2016).
5. Larsson, S.L., Skogsberg, J. & Bjorkegren, J. The low density lipoprotein receptor prevents secretion of dense apoB100-containing lipoproteins from the liver. *J Biol Chem* **279**, 831-6 (2004).
6. Rashid, S. *et al.* Decreased plasma cholesterol and hypersensitivity to statins in mice lacking Pcsk9. *Proc Natl Acad Sci U S A* **102**, 5374-9 (2005).
7. Eisele, B., Budzinski, R., Muller, P., Maier, R. & Mark, M. Effects of a novel 2,3-oxidosqualene cyclase inhibitor on cholesterol biosynthesis and lipid metabolism in vivo. *J Lipid Res* **38**, 564-75 (1997).
8. Soyninen, P., Kangas, A.J., Wurtz, P., Suna, T. & Ala-Korpela, M. Quantitative serum nuclear magnetic resonance metabolomics in cardiovascular epidemiology and genetics. *Circ Cardiovasc Genet* **8**, 192-206 (2015).
9. Wurtz, P. *et al.* Metabolite profiling and cardiovascular event risk: a prospective study of 3 population-based cohorts. *Circulation* **131**, 774-85 (2015).
10. Wurtz, P. *et al.* Quantitative Serum Nuclear Magnetic Resonance Metabolomics in Large-Scale Epidemiology: A Primer on -Omic Technologies. *Am J Epidemiol* **186**, 1084-1096 (2017).
11. Franzen, O. *et al.* Cardiometabolic risk loci share downstream cis- and trans-gene regulation across tissues and diseases. *Science* **353**, 827-30 (2016).
12. Swerdlow, D.I. *et al.* HMG-coenzyme A reductase inhibition, type 2 diabetes, and bodyweight: evidence from genetic analysis and randomised trials. *Lancet* **385**, 351-61 (2015).
13. Lee, S. *et al.* Network analyses identify liver-specific targets for treating liver diseases. *Mol Syst Biol* **13**, 938 (2017).
14. Mori, M. *et al.* Lanosterol synthase mutations cause cholesterol deficiency-associated cataracts in the Shumiya cataract rat. *J Clin Invest* **116**, 395-404 (2006).
15. Cohen, J. *Statistical power analysis for the behavioral sciences*, xxi, 567 p. (L. Erlbaum Associates, Hillsdale, N.J., 1988).
16. Zhu, J. *et al.* Increasing the power to detect causal associations by combining genotypic and expression data in segregating populations. *PLoS Comput Biol* **3**, e69 (2007).
17. Zhu, J. *et al.* Integrating large-scale functional genomic data to dissect the complexity of yeast regulatory networks. *Nat Genet* **40**, 854-61 (2008).
18. Tu, Z. *et al.* Integrating siRNA and protein-protein interaction data to identify an expanded insulin signaling network. *Genome Res* **19**, 1057-67 (2009).
19. Cohain, A. *et al.* Exploring the Reproducibility of Probabilistic Causal Molecular Network Models. *Pac Symp Biocomput* **22**, 120-131 (2017).

REVIEWERS' COMMENTS

Reviewer #1 (Remarks to the Author):

Revisions fully OK - well done!

Reviewer #3 (Remarks to the Author):

The authors did an excellent work during the revision of the paper. I recommend the publication of the paper in its current form.

Reviewer #4 (Remarks to the Author):

The Authors addressed most of my previous comments and provided methodological details and data as requested. However, there are still critical issues that remain to be addressed, as detailed below.

On a general note, in multiple instances the Authors seem to misperceive statistical significance of a correlation ($P < 0.05$) for the presence of a meaningful biological effect (i.e., a meaningful association between variation in gene expression and a given trait). Alongside r (correlation coefficient), I suggest reporting all correlations in terms of R^2 (coefficient for determination), which is more interpretable measure of association, and after assessing the correlations, all the weak correlation data to be removed from the manuscript. Please refer to Interpretation of the Correlation Coefficient: A Basic Review by Richard Taylor (1990) Research Article. <https://doi.org/10.1177/875647939000600106>, for some additional explanations on this matter.

Critical points:

About data reported in Figure 2C, the Authors reply "We first calculated the 1st principal component of each module as a proxy for the group of genes in that module and correlated PC 1 to all phenotype traits and corrected for multiple testing using BH method ($FDR < 0.05$). While we understand the concern expressed here and elsewhere about correlations in the range of 0.1-0.3 seeming weak, all correlations we report are highly significant after multiple test correction." Statistical significance does not mean or translates to meaningful biological effect. The fact that these weak correlations are statistically significant is most likely due to the number of data points used to calculate the correlations. In summary, the Authors should avoid over-interpreting these associations on the basis that "correlations we report are highly significant after multiple test correction", and (1) must formally acknowledge this limitation in the main text, recognizing that the identified correlations are weak in magnitude and if possible, in Figure 2 report R^2 (coefficient of determination) as a measure to express the strength of the associations (rather than using r) and (2) use this correlation analysis only to prioritize modules for following analyses. See also next point.

About weak data reported in Figure 3A (for which I have previously asked to provided additional support and/or replication) the Authors reply "The metabolite correlations are harder to independently verify as we don't have access to other datasets with gene expression and detailed metabolomics. We do note that while the correlations are not very strong, they are highly statistically significant and that the metabolomics correlations replicate the findings in STARNET liver and the morbidly obese cohort both in the broad categories of correlation but also their directionality."

As detailed earlier, this specific analysis reports extremely weak correlations. Some of these correlations are as low as $R^2 < 3\%$ (for Glucose) or $R^2 < 1\%$ (for HDL), and hence these cannot be deemed as biologically significant - regardless weather the respective p-value is less than 0.05.

These data should be removed from the manuscript as the reported biological associations are meaningless and of difficult interpretation. These data make the manuscript weaker and without validation/replication should not be reported in Nat Comm.

“Consistent with observations from STARNET, correlating the first principal component of this liver module with clinical metabolic traits measured in these 600 individuals found positive correlations with glucose levels and a negative correlation with plasma LDL levels (FDR < 5%) (Supplementary Table 10).”

Again, the Authors misperceive statistical significance for a meaningful biological effect. These correlations are $r = 0.13$ (glucose) and $r = -0.16489$ (plasma LDL). The respective R^2 are 1.7% and 2.7% meaning that the variability in module expression (1st PC, accounting only for fraction of the variability) explain less than 3% of the variability in the traits. These correlations are not biologically meaningful and given the potential for any environmental factor affecting the variability of glucose and plasma LDL levels, these data are too weak and of difficult biological interpretation to be reported without additional support. Perhaps I can suggest the Authors to assess the correlations between individual module genes (rather than the 1st PC) and these traits and combine (meta-analyze) the correlation results at the module level.

“Looking at the liver expression values for these 2 sets of HMDP mice, we found that the GLD gene mouse orthologues were positively correlated with blood glucose and some genes that were negatively correlated with cholesterol (Supplementary Fig. 5)”. I suggest looking at and reporting R^2 rather than r . These correlations need to be reported in a (Suppl.) Table too and, beyond the p -value for statistical significance, interpreted based on their biological “effect size” (here, strength of association).

“The first principal component of GLD module gene expression was 169 significantly correlated with 113 metabolites (FDR <5%, Supplementary Table 8, Supplementary Fig. 4), including a positive correlation with Glucose (Glc) and negative correlations with the cholesterol ester component of many lipoprotein size fractions”. It is not clear where the FDR of the correlation is reported in the Table as the columns heading of the table are not properly explained. Of these 113 “significant” correlations, only 4 account for 10% of the variation in the trait (and at least half of these correlations have a $R^2 < 5\%$). The positive correlation ($r=0.165$) with Glucose (Glc) accounts for <2.7% of the variation. Thus, the 1st PC of the GLD module (which explains 57% of the variation in gene expression of the module) explains 2.7% of the variation in glucose metabolite. What is the biological meaning of this correlation?

Reviewer: Enrico Petretto, PhD
Duke-NUS Medical School (Singapore)
Imperial College London (UK)

Reviewer #5 (Remarks to the Author):

To report about significance of a correlation the P -value of the T -test (with null hypothesis $r=0$) is enough and valid. Even, if the value of the correlation is low, it can be significant. This is due to the variability of the data used for correlation. I would therefore agree that reporting this as significant after multiple testing is correct.

If the authors state that the correlations are low but significant, this is correct.

The discussion if the findings are meaningful at the biological level is another one. In any case there is scientific and statistical evidence that variables are related linearly and it can be published as this.

I do not think that R^2 suggested by the reviewer (the coefficient of determination) is a better measure because it is used to quantify the adjustment to a linear regression. It solves a different

question, although also related to linear relationships this would, for me be very confusing. In conclusion, in my opinion: the statistical methods are correct, but I also agree that low correlations are difficult to interpret and therefore would suggest to be more careful with conclusions. I think that "highly significant" is not necessary. Just "significant" would be better.

Reviewer #1 (Remarks to the Author):

Revisions fully OK - well done!

Authors Response: We thank the reviewer for the helpful comments, and are glad we have satisfied them.

Reviewer #3 (Remarks to the Author):

The authors did an excellent work during the revision of the paper. I recommend the publication of the paper in its current form.

Authors Response: We thank the reviewer for the helpful comments, and we appreciate the compliment on our response.

Reviewer #4 (Remarks to the Author):

The Authors addressed most of my previous comments and provided methodological details and data as requested. However, there are still critical issues that remain to be addressed, as detailed below.

On a general note, in multiple instances the Authors seem to misperceive statistical significance of a correlation ($P < 0.05$) for the presence of a meaningful biological effect (i.e., a meaningful association between variation in gene expression and a given trait). Alongside r (correlation coefficient), I suggest reporting all correlations in terms of R^2 (coefficient for determination), which is more interpretable measure of association, and after assessing the correlations, all the weak correlation data to be removed from the manuscript. Please refer to Interpretation of the Correlation Coefficient: A Basic Review by Richard Taylor (1990) Research Article. <https://doi.org/10.1177/875647939000600106>, for some additional explanations on this matter.

Authors Response: As Reviewer #5 points out below, a statistically significant p-value for correlation analysis indicates that there is evidence of a relationship, regardless of the strength of that relationship. Transcriptomic and genomic analyses regularly report associations that are numerically weak but highly statistically significant. GWAS in particular routinely report as biologically significant associations that are many orders of magnitude smaller than those we report here, and publish these findings in high-impact journals. Furthermore, the claims of biological significance for the GLD module we make in the manuscript are supported by multiple different lines of evidence, including existing annotations of genes and *in vivo* experimentation, and do not rest entirely or even primarily on these correlation analyses.

We believe that these claims are entirely justified, and that it is entirely appropriate to present this correlation analysis as one piece of evidence towards these claims.

We also agree with Reviewer #5 below that r^2 is a different quantity that measures a different feature of the correlation, and presenting it would be confusing.

Critical points:

About data reported in Figure 2C, the Authors reply ???We first calculated the 1st principal component of each module as a proxy for the group of genes in that module and correlated PC1 to all phenotype traits and corrected for multiple testing using BH method (FDR < 0.05). While we understand the concern expressed here and elsewhere about correlations in the range of 0.1-0.3 seeming weak, all correlations we report are highly significant after multiple test correction.???

Statistical significance does not mean or translates to meaningful biological effect. The fact that these weak correlations are statistically significant is most likely due to the number of data points used to calculate the correlations. In summary, the Authors should avoid over-interpreting these associations on the basis that ???correlations we report are highly significant after multiple test correction???, and (1) must formally acknowledge this limitation in the main text, recognizing that the identified correlations are weak in magnitude and if possible, in Figure 2 report R^2 (coefficient of determination) as a measure to express the strength of the associations (rather than using r) and (2) use this correlation analysis only to prioritize modules for following analyses. See also next point.

Authors Response: We have edited the portion of the text referring to this analysis to read:

“16 of these modules had statistically significant correlations of any magnitude with plasma lipid and blood glucose levels when considering the first principal component of gene expression in each module (mean variation explained: 52%, range: 42-64%, **Supplementary Table 1**). Similar but weaker correlations were observed in higher principal components (**Supplementary Data 4**).

Though no principal component of any module showed overwhelmingly strong correlations with lipid or glucose traits, only one module in the liver (hereafter, the Glucose and Lipid Determining module; GLD) had any principal component that was negatively correlated with lipid traits (LDL and plasma cholesterol levels) and positively correlated with glucose traits (HbA1c and blood glucose levels) (FDR < 5%, **Fig. 2**).”

About weak data reported in Figure 3A (for which I have previously asked to provide additional support and/or replication) the Authors reply ???The metabolite correlations are harder to independently verify as we don???t have access to other datasets with gene expression and detailed metabolomics. We do note that while the correlations are not very strong, they are highly statistically significant and that the metabolomics correlations replicate the findings in STARNET liver and the morbidly obese cohort both in the broad categories of correlation but also their directionality.???

As detailed earlier, this specific analysis reports extremely weak correlations. Some of these correlations are as low as $R^2 < 3\%$ (for Glucose) or $R^2 < 1\%$ (for HDL), and hence these cannot be deemed as

biologically significant - regardless whether the respective p-value is less than 0.05. These data should be removed from the manuscript as the reported biological associations are meaningless and of difficult interpretation. These data make the manuscript weaker and without validation/replication should not be reported in Nat Comm.

Authors Response: We have edited the portion of the text referring to this analysis to read:

“The first principal component of GLD module gene expression had weak but significant correlations with 113 metabolites (FDR <5%, **Supplementary Data 7, Supplementary Fig. 4**), including a positive correlation with Glucose (Glc) and negative correlations with the cholesterol ester component of many lipoprotein size fractions (VLDL: XS, S, M, L, XXL, LDL: S, M, L, HDL: S, M, L, and IDL), consistent with the associations between this module and the glucose and lipid clinical traits (**Fig. 3A**).”

We disagree with the reviewer’s argument that this analysis is unsupported, as it is only presented in the context of being consistent with the previously reported analyses. It is entirely consistent with all the other data we present, and we do not believe it is reasonable to require further replication beyond what is already present in the manuscript, nor to require its removal from the manuscript.

???Consistent with observations from STARNET, correlating the first principal component of this liver module with clinical metabolic traits measured in these 600 individuals found positive correlations with glucose levels and a negative correlation with plasma LDL levels (FDR < 5%) (Supplementary Table 10).???

Again, the Authors misperceive statistical significance for a meaningful biological effect. These correlations are $r = 0.13$ (glucose) and $r = -0.16489$ (plasma LDL). The respective R^2 are 1.7% and 2.7% meaning that the variability in module expression (1st PC, accounting only for fraction of the variability) explain less than 3% of the variability in the traits. These correlations are not biologically meaningful and given the potential for any environmental factor affecting the variability of glucose and plasma LDL levels, these data are too weak and of difficult biological interpretation to be reported without additional support. Perhaps I can suggest the Authors to assess the correlations between individual module genes (rather than the 1st PC) and these traits and combine (meta-analyze) the correlation results at the module level.

Authors Response: We have edited the quoted sentence to clarify that we are referring to statistical significance and not making claims about biological significance based on this one analysis:

“Consistent with observations from STARNET, correlating the first principal component of this liver module with clinical metabolic traits measured in these 600 individuals found statistically significant positive correlations with glucose levels and a statistically significant negative correlation with plasma LDL levels (FDR < 5%) (**Supplementary Table 2**).”

However, we disagree with the reviewer’s characterization of these r^2 values as extremely small. A 2016 study estimating the heritability explained by common genetic variation in various human traits estimated that all genetic effects combined explained only 22% of variation in blood glucose and 33% of variation in plasma LDL, and all loci found by GWAS combined explained only 2.6% of variation in blood glucose and 7.8% of variation in plasma LDL (<https://doi.org/10.1016/j.ajhg.2016.05.013>, Table 1). Thus,

the reviewer is estimating that the variability explained by the first principal component of the GLD module is equivalent to about 5-10% of genetic variability in these traits, representing an effect of a similar order of magnitude to that of all loci discovered by GWAS combined. The fact that environmental effects account for more than this does not invalidate the entire field of GWAS, and it does not invalidate this analysis.

The suggestion of meta-analyzing individual GLD member gene correlations is an interesting one. It would not be trivial to implement, as standard meta-analysis combines multiple measurements of the same association, not measurements of different associations that are expected to have an aggregate effect. The effects of different genes in the module may be different magnitudes or different directions without detracting from the claim that the entire module has one effect. This analysis would therefore require implementation of a completely novel method, which is outside the scope of this work.

Looking at the liver expression values for these 2 sets of HMDP mice, we found that the GLD gene mouse orthologues were positively correlated with blood glucose and some genes that were negatively correlated with cholesterol (Supplementary Fig. 5). I suggest looking at and reporting R^2 rather than r . These correlations need to be reported in a (Suppl.) Table too and, beyond the p-value for statistical significance, interpreted based on their biological effect size (here, strength of association).

Authors Response: Again, we agree with Reviewer #5 that r^2 is a different quantity and it is not appropriate to report it here.

The first principal component of GLD module gene expression was 169 significantly correlated with 113 metabolites (FDR <5%, Supplementary Table 8, Supplementary Fig. 4), including a positive correlation with Glucose (Glc) and negative correlations with the cholesterol ester component of many lipoprotein size fractions. It is not clear where the FDR of the correlation is reported in the Table as the columns heading of the table are not properly explained. Of these 113 significant correlations, only 4 account for 10% of the variation in the trait (and at least half of these correlations have a R^2 <5%). The positive correlation ($r=0.165$) with Glucose (Glc) accounts for <2.7% of the variation. Thus, the 1st PC of the GLD module (which explains 57% of the variation in gene expression of the module) explains 2.7% of the variation in glucose metabolite. What is the biological meaning of this correlation?

Authors Response: See above for the edit we have made to this sentence. We have also added a description of the statistical tests used to the caption of this supplementary data file. The FDR is reported in the column labeled "FDR." Again, 2.7% of variation in glucose is actually very large in the context of previously reported genetic effects, and the idea that it is too insignificant to be reported is absurd.

Reviewer: Enrico Petretto, PhD

Duke-NUS Medical School (Singapore)

Imperial College London (UK)

Reviewer #5 (Remarks to the Author):

To report about significance of a correlation the P-value of the T-test (with null hypothesis $r=0$) is enough and valid. Even, if the value of the correlation is low, it can be significant. This is due to the variability of the data used for correlation. I would therefore agree that reporting this as significant after multiple testing is correct.

If the authors state that the correlations are low but significant, this is correct.

The discussion if the findings are meaningful at the biological level is another one. In any case there is scientific and statistical evidence that variables are related linearly and it can be published as this.

I do not think that R^2 suggested by the reviewer (the coefficient of determination) is a better measure because it is used to quantify the adjustment to a linear regression. It solves a different question, although also related to linear relationships this would, for me be very confusing.

In conclusion, in my opinion: the statistical methods are correct, but I also agree that low correlations are difficult to interpret and therefore would suggest to be more careful with conclusions. I think that "highly significant" is not necessary. Just "significant" would be better.

Authors Response: We thank the reviewer for this opinion, which we agree with. We would like to clarify that the quote of "highly significant" used by Reviewer #4 above is from our response to the previous round of peer review, not from the text itself. It does appear at one point in the text in reference to a different analysis, and we have removed it. The claims of significance for the analyses in question in the current version of the manuscript are quoted in our responses to Reviewer #4 above.